# KSHV-encoded vCyclin can modulate HIF1α levels to promote DNA replication in hypoxia

**Rajnish Kumar Singh[1,2], Yonggang Pei[1], Dipayan Bose[1], Zachary L Lamplugh[1], Kunfeng Sun[1], Yan Yuan[3], Paul Lieberman[4], Jianxin You[5], Erle S Robertson[1]\***

[1]Department of Otorhinolaryngology-Head and Neck Surgery, Perelman School of Medicine, University of Pennsylvania, Philadelphia, United States; [2]Institute of Medical Sciences, Banaras Hindu University, Varanasi, India; [3]Department of Microbiology, Levy Building, School of Dental Medicine, University of Pennsylvania, Philadelphia, United States; [4]Program in Gene Regulation, The Wistar Institute, Philadelphia, United States; [5]Department of Microbiology, Perelman School of Medicine, University of Pennsylvania, Philadelphia, United States

**Abstract** The cellular adaptive response to hypoxia, mediated by high HIF1α levels includes metabolic reprogramming, restricted DNA replication and cell division. In contrast to healthy cells, the genome of cancer cells, and Kaposi's sarcoma associated herpesvirus (KSHV) infected cells maintains replication in hypoxia. We show that KSHV infection, despite promoting expression of HIF1α in normoxia, can also restrict transcriptional activity, and promoted its degradation in hypoxia. KSHV-encoded vCyclin, expressed in hypoxia, mediated HIF1α cytosolic translocation, and its degradation through a non-canonical lysosomal pathway. Attenuation of HIF1α levels by vCyclin allowed cells to bypass the block to DNA replication and cell proliferation in hypoxia. These results demonstrated that KSHV utilizes a unique strategy to balance HIF1α levels to overcome replication arrest and induction of the oncogenic phenotype, which are dependent on the levels of oxygen in the microenvironment.

**\*For correspondence:**
erle@pennmedicine.upenn.edu

**Competing interests:** The authors declare that no competing interests exist.

## Introduction

Kaposi's sarcoma associated herpesvirus is the causative agent of Kaposi's sarcoma (KS) and is tightly linked to Primary effusion lymphoma (PEL) and Multicentric Castleman disease (MCD) (**Boshoff and Weiss, 2002**; **Goncalves et al., 2017**; **Chang et al., 1994**). Evidence also suggests that there is a strong association between KSHV infection and KSHV-associated inflammatory cytokine syndrome (KICS) (**Cantos et al., 2017**; **Polizzotto et al., 2012**). The complete nucleotide sequence of KSHV show long unique regions (LURs) that encodes approximately 90 open reading frames (**Russo et al., 1996**). Many KSHV-encoded genes are also identified as homologs of cellular genes such as vCyclin, vFLIP, vGPCR, and vIRFs (**Mesri et al., 2014**; **Giffin and Damania, 2014**). Upon successful infection of host cells, the virus chooses to either enter a latent phase or continue toward lytic replication to generate more copies of infectious virions (**Chandran, 2010**; **Nicol et al., 2016**). To establish latency, the viral genome transitions through a series of epigenetic modifications that are predominantly at lysine residues on histones to minimize expression of the majority of encoded genes. Only a fraction of genes critical for latency is expressed (**Uppal et al., 2015**; **Toth et al., 2010**).

The major KSHV-encoded genes expressed in the majority of KSHV-positive cancer biopsy samples include the latency-associated nuclear antigen (LANA), the virus encoded homolog of human cyclin D (vCyclin), the homolog of FLICE-like inhibitory protein (vFLIP) and homologs of Interferon

regulatory factors (vIRF1, vIRF2, vIRF3, and vIRF4) (*Arias et al., 2014*; *Damania, 2004*). Additionally, KSHV-positive biopsy samples are also known to have transcripts for the human homolog of the G-protein-coupled receptor (vGPCR) and glycoprotein K8 (*Arias et al., 2014*; *Damania, 2004*). KSHV-encoded LANA plays an essential role in maintenance of KSHV episomes within the host by tethering the KSHV DNA to the host genome (*Uppal et al., 2014*; *Cotter and Robertson, 1999*; *Ballestas et al., 1999*). LANA-mediated KSHV genome persistence requires its C-terminal, which binds to the KSHV DNA within its terminal repeat region while the N-terminal region of LANA binds with the host nuclear DNA (*Schwam et al., 2000*). LANA, in addition to tethering of the KSHV genome regulates several cellular functions required for oncogenic transformation of infected cells. The major pathways modulated by KSHV-encoded LANA include cell cycle (*Wei et al., 2016*), apoptosis (*Friborg et al., 1999*), Epithelial-Mesenchymal transition (EMT) (*Gasperini et al., 2012*; *Jha et al., 2016*), and chromosome instability (CIN) (*Lang et al., 2018*). Nevertheless, other KSHV-encoded latent genes also play a major role in oncogenic transformation of infected cells. KSHV-encoded vCyclin is a homolog of host Cyclin D and is known to facilitate the G1/S transition and override contact inhibition of viral-infected cells (*Jones et al., 2014*; *Verschuren et al., 2002*). Another latent antigen vFLIP is known to target the NF-kB pathway to promote oncogenesis (*Sun and Cesarman, 2011*; *Zhao et al., 2015*; *Rahman and McFadden, 2011*). Similarly, vIRFs can inhibit pro-apoptotic pathways such as those mediated by the BH3-family of proteins (*Choi et al., 2012*). KSHV-encoded vGPCR, considered to be a lytic gene, was also shown to be expressed in a number of KSHV positive-tumor biopsy samples (*Damania, 2004*). It was shown to be a bonafide oncoprotein, that regulates activities of the MAPK pathway through p38 and upregulation of the hypoxia survival protein HIF1α (*Montaner et al., 2013*; *Vischer et al., 2014*).

The latent to lytic switch of KSHV in infected cells is still not yet fully understood, but the regulation of epigenetic programs are known to be critical for this switch. Hence, the use of 12-*O*-tetradecanoylphorbol-13-acetate (TPA) and Butyric acid (BA), potent epigenetic modifiers, are effective inducers of in-vitro reactivation of KSHV from latently-infected cells. In addition to epigenetic modulation by TPA/BA, exposure of KSHV positive cells to hypoxia or reactive oxygen species can induce KSHV reactivation (*Ye et al., 2011a*; *Aneja and Yuan, 2017*; *Davis et al., 2001*; *Cai et al., 2006a*; *Ye et al., 2011b*). Independent of the method utilized for KSHV reactivation, replication competency and favorable conditions are required for initiation, as well as progression of DNA replication. Among the known mediators of KSHV lytic replication, the mechanism of reactivation in response to hypoxia is still not fully understood because of its overall negative effect on replication, transcription, translation, or energy production.

Hypoxia is detrimental to aerobic cells depriving them of energy supplies due to the dramatic drop in their ability to generate ATP through oxidative phosphorylation. The adaptive response to hypoxia includes arrest of cell cycle, mainly at G1/S transition, and DNA replication due to a loss of available cellular energy stores, and the macromolecular complexes that drive these processes (*Mason and Rathmell, 2011*; *Michiels, 2004*). In hypoxia, the specialized transcription factors (Hypoxia inducible factors; HIFs) are stabilized to activate transcription of stress-associated genes responsible for metabolic reprogramming, survival, angiogenesis, and anti-apoptosis (*Dengler et al., 2014*; *Majmundar et al., 2010*). Nevertheless, stabilization of HIF1α is known for arresting G1/S transition through regulation of the cyclin dependent kinase inhibitors p21 and p27, and other related pathways (*Gardner et al., 2001*; *Hammond et al., 2002*; *Goda et al., 2003*).

KSHV, like other oncogenic viruses, utilizes multiple mechanisms to maintain a high basal level of HIF1α in infected cells and infected cells are adjusted to survive with higher HIF1α levels as evident from their normal proliferation and replication potential. Paradoxically, the KSHV genome in infected cells undergoes enhanced lytic replication upon further induction of hypoxia through their ability to bypass HIF1α-mediated block to DNA replication.

The aim of the present study is to investigate the mechanism by which KSHV bypasses HIF1α-mediated repression of replication of infected cells during hypoxia. We show that, despite elevated levels of HIF1α protein, the presence of KSHV can also restrict HIF1α activity during hypoxia. Specifically, KSHV-encoded vCyclin expressed in hypoxia physically interacts with HIF1α and abrogates its transcriptional activity by mediating its degradation through a non-canonical lysosomal pathway. Knock-down of vCyclin resulted in a compromised potential of KSHV-positive cells to bypass hypoxia-mediated suppression of DNA replication, as well as their potential to proliferate in an anchorage-dependent or independent manner. These results provide new evidence to a stringently

regulated control of HIF1α levels by two of its critical latent antigens, important for driving the oncogenic phenotype of KSHV positive cells in hypoxia.

## Results

### KSHV Infection promotes DNA replication in hypoxia by interfering with HIF1α signaling

Elevated HIF1α levels restrict DNA replication through multiple pathways. KSHV infected cells show the signature of a hypoxic microenvironment with elevated levels of HIF1α, although they replicate and divide spontaneously (*Cuninghame et al., 2014*). Further, introduction of external hypoxia to KSHV positive cells is known to reactivate the virus resulting in induction of lytic replication (*Davis et al., 2001*; *Cai et al., 2006a*). We hypothesized that despite increased levels of HIF1α in KSHV-infected cells, the virus may negatively regulate HIF1α-mediated suppression of replication to bypass hypoxia-induced arrest of DNA replication. To investigate, we used BJAB cells containing BAC-KSHV (BJAB-KSHV cells) and the KSHV-negative BJAB cells as the isogenic cell control (with similar passage number), grown either in normoxic or hypoxic conditions. BJAB-KSHV cells were previously analyzed for other characteristics of latently infected KSHV-positive cells (*Singh et al., 2018*). Analysis of replication efficiency through measuring nucleotide analog incorporation in these cell lines indicated that the presence of KSHV supported sustained DNA replication in hypoxic conditions (*Figure 1A,B*). As HIF1α is one of the major effectors of hypoxia and mediates repression of DNA replication in hypoxic conditions, we investigated the protein levels of HIF1α in BJAB/BJAB-KSHV cells grown in hypoxia (*Figure 1C*). Western blot analysis suggested an unexpected pattern of HIF1α levels in BJAB-KSHV cells which were dramatically reduced at the 36 hr time point (*Figure 1C*, lower panel). The level of HIF1α remained high in BJAB-KSHV cells as compared to BJAB cells except at this single time point of 36 hr (*Figure 1C*).

To further corroborate the above findings, we examined the status of known transcriptional targets of HIF1α at 36 hr time point. The lactate dehydrogenase subunit A (LDHA), Prolyl 4-hydroxylase subunit alpha-1(P4HA1), and the Pyruvate Dehydrogenase Kinase 1 (PDK1) were examined as they are known to be regulated by HIF1α. The results showed that the presence of KSHV had a restricted transcriptional profile for HIF1α in hypoxic conditions (*Figure 1D*). For cells grown under normoxic conditions, the fold change in expression of all the studied HIF1α targets showed a significant elevated level of expression in BJAB-KSHV cells as compared to BJAB cells (2.4-fold for LDHA, 3.9-fold for P4HA1 and 1.6-fold for PDK1), suggesting a relatively high level of endogenous HIF1α in these cells (*Figure 1D*). To confirm this high level of endogenous HIF1α in BJAB-KSHV cells, another time point (12 hr) was used to perform real-time PCR using PDK1 as a representative candidate. The results also showed a high level of HIF1α at this time point (*Figure 1—figure supplement 1*). Interestingly, when the expression of these targets were determined in either BJAB or BJAB-KSHV cells grown in hypoxia compared to cells from the matched experiments grown under normoxic conditions, the hypoxic BJAB-KSHV cells showed a compromised up-regulated expression as compared to BJAB cells in hypoxia (*Figure 1D*). Specifically, expression of LDHA, P4HA1, and PDK1 in hypoxic BJAB cells showed up-regulation of 2.5, 7.0 and 2.8-fold change, respectively, when compared to normoxic BJAB cells. However, expression of the same targets in hypoxic BJAB-KSHV cells showed up-regulation of only 0.85, 2.47 and 1.45-fold change, respectively, compared to normoxic BJAB-KSHV cells (*Figure 1D*). To further corroborate these results, we checked the expression of these HIF1α targets in HEK293T and HEK293T-BAC16-KSHV cells grown under normoxic or hypoxic conditions. A similar pattern of expression was also observed in these cells, suggesting that KSHV infection may restrict HIF1α transcriptional activity under hypoxic conditions (*Figure 1—figure supplement 1*). Further validation of these results was done by infecting primary blood mononuclear cells with purified KSHV virions. PBMCs infection was carried out for 24 hr followed by incubation under normoxic or hypoxic conditions. A fraction of infected cells was used to confirm the efficiency of infection using LANA immunofluorescence (*Figure 1—figure supplement 1*). Analysis of the HIF1α target PDK1, in infected PBMCs grown under hypoxic conditions compared to uninfected cells, showed a similar result as observed for BJAB vs BJAB-KSHV, or HEK293T vs HEK293T-BAC-KSHV cells (*Figure 1—figure supplement 1*). This strongly suggest a role for KSHV in the negative regulation of HIF1α on infection of human cells in hypoxic conditions. Finally, to rule out any effect

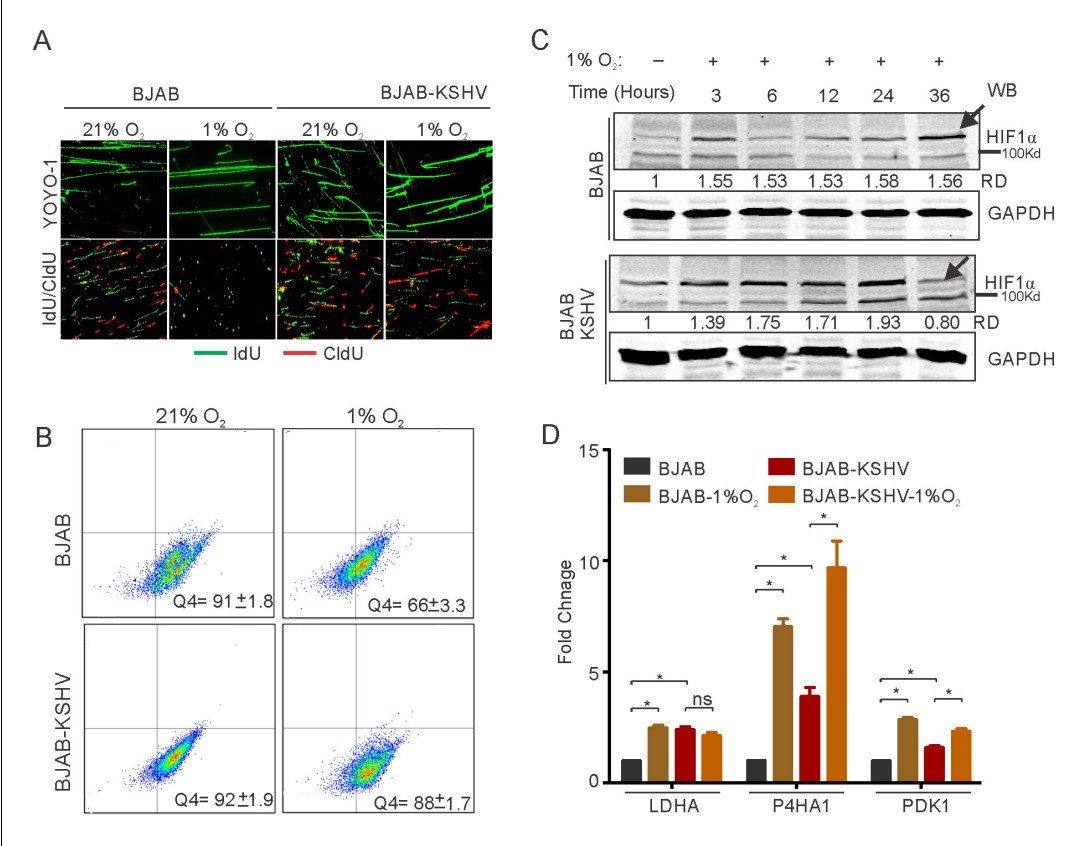

**Figure 1.** KSHV infection restricts HIF1α activity to promote DNA replication. (**A**) Representative image of YOYO-1 staining of stretched DNA and nucleotide analog incorporation in BJAB and BJAB-KSHV cells grown under normoxic or hypoxic conditions (**B**) quantitation of nucleotide analog incorporation in BJAB and BJAB-KSHV cells grown under normoxic or hypoxic conditions. Cells pulsed with IdU were probed with anti-IdU antibodies and the relative incorporation was measured using a FACS machine. (**C**) Representative HIF1α western blot analysis in BJAB and BJAB-KSHV cells grown under normoxia or hypoxic conditions. The cells were either grown under normoxic or hypoxic conditions for the indicated time period. Equal amounts of protein were used to detect HIF1α from the lysate of these cells. GAPDH western Blot served as the loading control. (**D**) Real-time PCR analysis for the transcriptional activity of HIF1α in BJAB and BJAB-KSHV cells grown under normoxic or hypoxic conditions. The cells were grown under normoxic or hypoxic conditions followed by total RNA isolation and cDNA synthesis. Equal amounts of cDNA were used for expression analysis of HIF1α targets (LDHA, PDK1, and P4HA1) by real-time PCR. The experiment was performed in triplicate. The error bar represents standard error from the mean. A p-value of <0.05 (*) was taken into consideration for statistical significance.

The online version of this article includes the following source data and figure supplement(s) for figure 1:

**Source data 1.** Fold change of LDHA P4HA1 and PDK1 in BJAB AND BJAB_KSHV cells in Normoxia and hypoxia.

**Figure supplement 1.** KSHV mediated supression of HIF1α transcriptional activity in HEK293T cells and in PBMCs.

**Figure supplement 1—source data 1.** Fold change of HIF1α transcriptional targets in various cell types in normoxic conditions.

of saturating endogenous levels of HIF1α in KSHV-positive cells on the observed restricted transcriptional activity, we performed real-time experiments at several time points taking PDK1 as a representative candidate in both BJAB and BJAB-KSHV cells. The results clearly showed a continuous increase in transcriptional activity of HIF1α in KSHV-negative BJAB cells (*Figure 1—figure supplement 1*). Interestingly, for BJAB-KSHV cells a sharp drop in transcriptional activity of HIF1α followed by an increase in its activity was observed (*Figure 1—figure supplement 1*). This clearly suggested a possible inverted regulation of HIF1α levels and activity by KSHV under hypoxic conditions.

## KSHV-encoded vCyclin restricts HIF1α transcriptional activity

To confirm the repression of HIF1α by KSHV in hypoxia and rule out possibilities that the less pronounced effect of HIF1α in BJAB-KSHV during hypoxia was due to the high level of HIF1α, we transfected HEK293T cells with KSHV-encoded genes which are known to be differentially expressed under hypoxic conditions (*Cai et al., 2006a*; *Singh et al., 2018*; *Veeranna et al., 2012*). The

expression of transfected KSHV-encoded genes was confirmed by Western blot analysis against Myc-tag (*Figure 2A*). The cells were grown either in normoxic or hypoxic conditions and HIF1α transcriptional activity was checked by real-time expression of the well-established targets of HIF1α; LDHA, P4HA1, and PDK1. The results clearly showed that expression of KSHV-encoded vCyclin led to a drastic reduction in transcriptional activity of HIF1α based on the levels of its downstream targets, under both normoxic and hypoxic conditions (*Figure 2B–D* and *Figure 2—figure supplement 1*). Briefly, a reduction of two- to fourfold in expression of LDHA, P4HA1, and PDK1 was observed in vCyclin expressing cells grown under hypoxic conditions as compared to the cells expressing mock or other KSHV-encoded genes including LANA and RTA grown under hypoxia (*Figure 2B–D*). Furthermore, we showed that the fold change in expressions of LDHA, P4HA1 and PDK1 in vCyclin

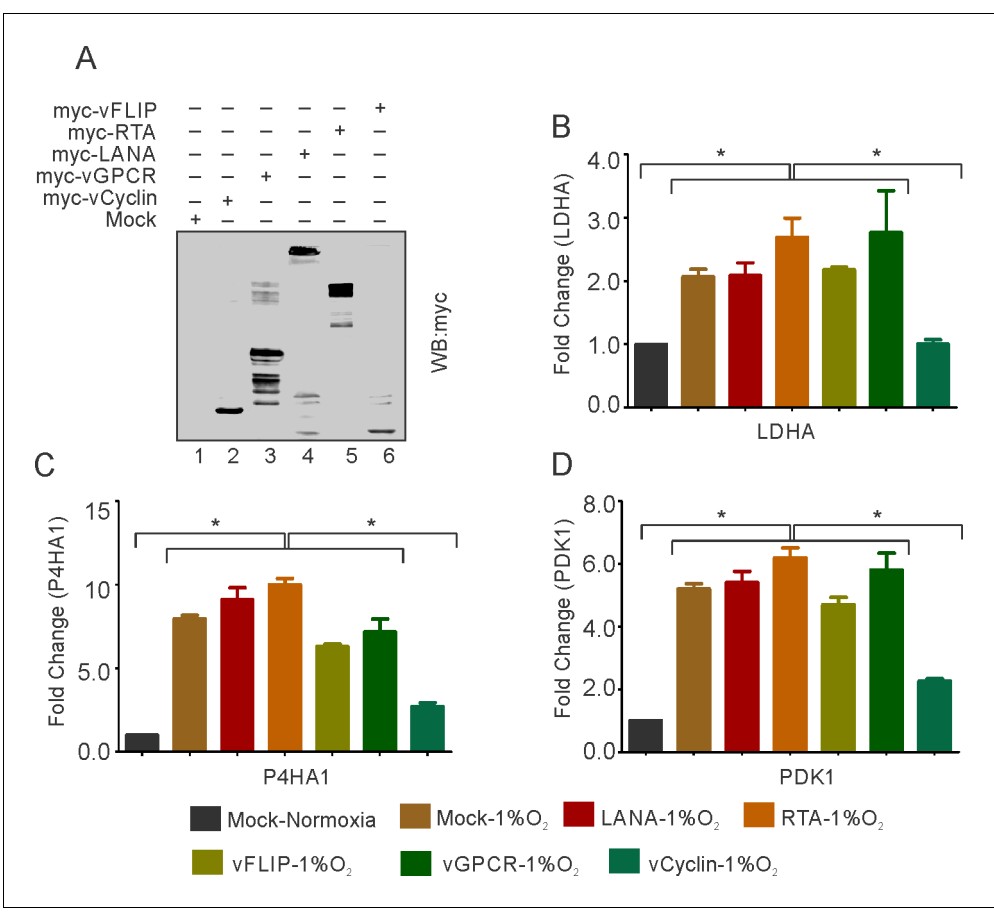

**Figure 2.** KSHV-encoded vCyclin restricts HIF1α transcriptional activity. (**A**) Expression of KSHV-encoded antigens. Mock, myc-tagged vCyclin, myc-tagged vGPCR, myc-tagged LANA, myc-tagged RTA, or myc-tagged vFLIP plasmids were transfected into HEK293T cells followed by western blot analysis using anti-myc antibody. (**B–D**) Real-time PCR expression analysis of HIF1α targets LDHA, P4HA1 and PDK1 in HEK293T cells transfected with Mock, LANA, RTA, vFLIP, vCyclin, or vGPCR plasmids and grown under hypoxic conditions compared to Mock transfection grown under normoxic or hypoxic conditions. Twenty-four hr post transfection, cells were grown for another 24 hr in hypoxic conditions. Total RNA was isolated from transfected cells followed by synthesis of cDNA using 2 μg of the total RNA. Equal cDNA was used for quantitative real-time PCR. GAPDH was used as the endogenous control. The experiments were done in triplicate. The error bar represents standard error from the mean. A p-value of <0.05 (*) was taken into consideration for statistical significance.

The online version of this article includes the following source data and figure supplement(s) for figure 2:

**Source data 1.** KSHV-encoded vCyclin mediated restriction of HIF1α transcriptional activity in hypoxic conditions.

**Figure supplement 1.** Real-time PCR expression analysis of HIF1α targets LDHA, P4HA1 and PDK1 in HEK293T cells transfected with Mock, LANA, vGPCR, RTA, vCyclin or vFLIP plasmids and grown under normoxic conditions.

**Figure supplement 1—source data 1.** KSHV-encoded vCyclin mediated restriction of HIF1α transcriptional activity in normoxic conditions.

expressing cells grown under normoxic conditions were also downregulated by 0.43, 0.83, and 0.50-fold when compared to mock transfected cells, respectively (*Figure 2—figure supplement 1*). Also, the KSHV-encoded vGPCR, a known positive regulator of HIF1α showed fold changes in expression of LDHA, P4HA1, and PDK1 in vGPCR transfected cells to be 1.43, 1.87, and 1.9 as compared to mock transfected cells, respectively (*Figure 2—figure supplement 1*).

## KSHV-encoded vCyclin associates with HIF1α to mediate its cytosolic translocation

We wanted to investigate the mechanism by which vCyclin negatively regulates HIF1α transcriptional activity. We first determined whether vCyclin can associate with HIF1α in a molecular complex to inhibit its transcriptional activity. We transfected HEK293T cells with GFP-vCyclin expression plasmid to monitor their interaction. Interestingly, initial investigation for the expression of GFP-vCyclin showed a variable localization of GFP signals in both nuclear and cytosolic compartments. In the initial 24 hr, the GFP signals were predominantly localized to nuclear compartments while at a later 48 hr time, localization to the outer cytosolic compartment as punctate signals were observed (*Figure 3—figure supplement 1*). To further support this nuclear to cytosolic translocation of expressed GFP-vCyclin, we performed high-resolution confocal microscopy to determine the localization of the expressed GFP-vCyclin. The results showed that vCyclin expression maintained a discrete pattern localized in the nucleus, cytoplasm or both nuclear and cytosolic compartments (*Figure 3A*). We hypothesized that vCyclin may associate directly with HIF1α to mediate its cytosolic translocation and hence reduce its transcriptional activity. To validate this hypothesis, we grew KSHV-positive BC3 cells in hypoxia and monitored the localization of both vCyclin and HIF1α. Colocalization of vCyclin and HIF1α in nuclear, cytosolic, or both nuclear and cytosolic compartments demonstrated a potential role for vCyclin in HIF1α translocation from a prominently nuclear to cytosolic compartment localization (*Figure 3B*). To support these data which showed the potential involvement of vCyclin in HIF1α translocation, we performed immuno-precipitation experiments using HIF1α specific antibodies. The results demonstrate that vCyclin can associate with HIF1α in molecular complexes in KSHV-positive cells (*Figure 3C*). We further validated the interaction between vCyclin and HIF1α in physiologically relevant, naturally infected KSHV positive cells (BC3, BCBL1, and JSC1), which were used in immuno-precipitation assays. BJAB, a KSHV negative cell line was also included as a control. The results clearly demonstrated that KSHV-encoded vCyclin interacts with HIF1α in KSHV-positive cells (*Figure 3D*). Western blot analysis for HIF1α on cytosolic or nuclear fractions of HEK293T cells with and without vCyclin expression further confirmed that vCyclin-mediated HIF1α cytosolic translocation (*Figure 3E*).

## KSHV-encoded vCyclin down-regulates HIF1α through the lysosomal pathway

We investigated the mechanism by which KSHV-encoded vCyclin inhibited HIF1α transcriptional activity. We questioned whether the repression of HIF1α by vCyclin was due to suppression of its transcriptional activity (based on physical interaction), or was it due to changes at the level of the protein (degradation), as we observed reduced expression of HIF1α in KSHV positive cells after 36 hr post-hypoxia induction (*Figure 1C*). We transfected HEK293T cells with an expression plasmid encoding vCyclin followed by western blot analysis for HIF1α. To rule out involvement of other KSHV antigens that was previously shown to be associated with HIF1α, we also transfected cells with KSHV-encoded LANA, vGPCR, vFLIP, and RTA plasmids (*Figure 4A*). Expression of KSHV-encoded antigens was previously confirmed by western blots against the fused epitope tag or the viral antigen (*Figure 2A*), followed by monitoring expression of HIF1α (*Figure 4A*). The results demonstrated that expression of vCyclin significantly down-regulated HIF1α at the protein level (*Figure 4A*). To further corroborate these findings, we transfected HEK293T cells with increasing amounts of expression plasmids encoding vCyclin followed by western blot to monitor HIF1α expression. As expected, HIF1α expression levels were continually decreased with increasing amounts of vCyclin (*Figure 4B*). These results were supported using another cell line Saos-2 (*Figure 4—figure supplement 1*). To further study the detailed mechanism of vCyclin-mediated degradation of HIF1α, we generated stable cells with mock or GFP-vCyclin expressing plasmids. Interestingly, we observed that in contrast to mock transfected cells, signals for vCyclin as monitored by GFP gradually decreased and

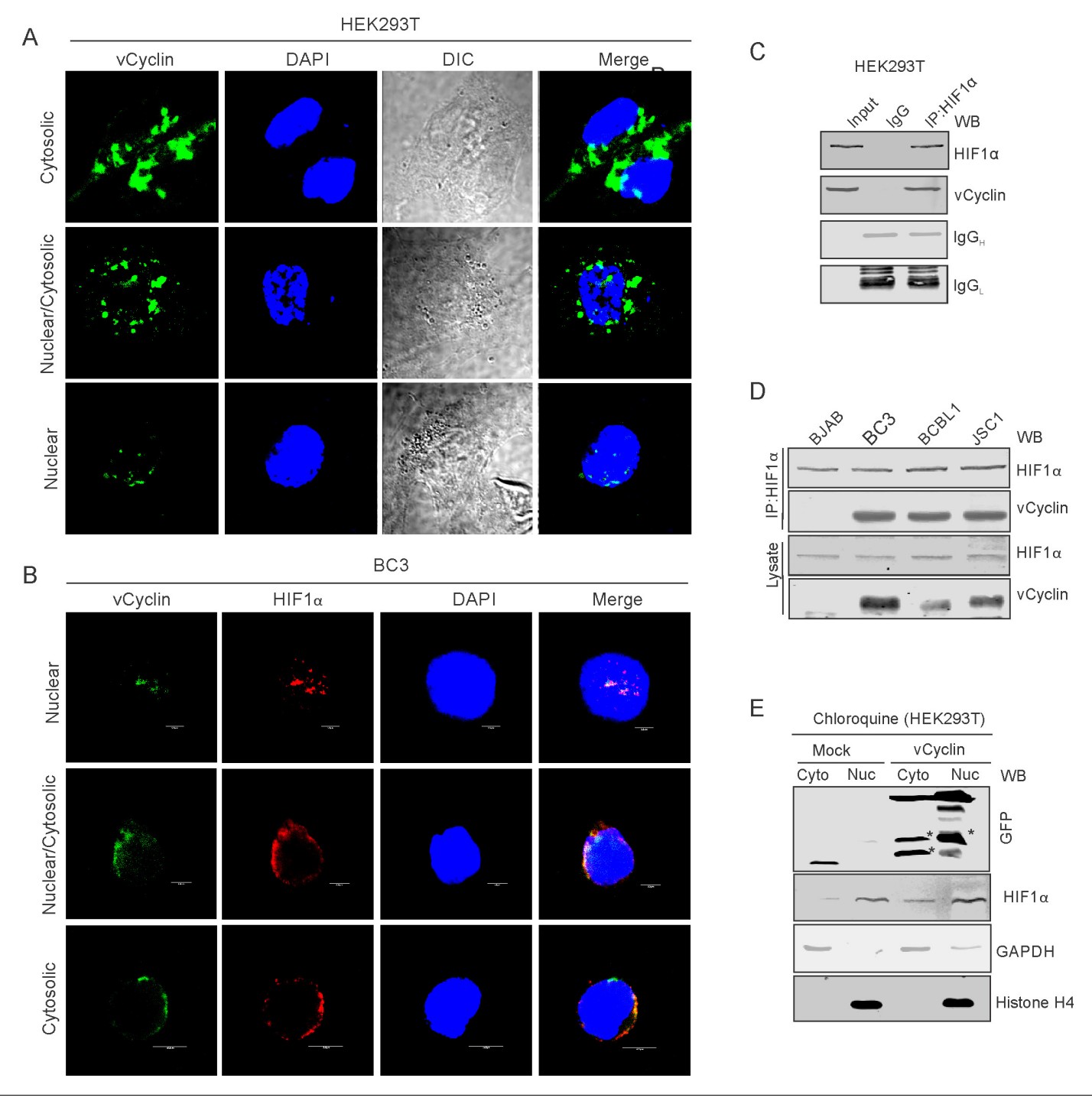

**Figure 3.** KSHV-encoded vCyclin physically interacts with HIF1α and mediates its cytosolic translocation. (**A**) HEK293T cells were transfected with GFP-vCyclin encoding plasmids. The transfected cell were analyzed for cellular localization of expressed proteins through GFP fluorescence with reference to nucleus (DAPI signals). The GFP signals showed a clear distribution to the nuclear, cytosolic or nucleo-cytosolic compartments. (**B**) Microscopic investigation of HIF1α sub-cellular localization in vCyclin expressing KSHV-positive BC3 cells grown under hypoxic conditions. Cells were seeded on coverslips. The coverslip-attached cells were fixed, permealized, blocked and probed with vCyclin and HIF1α antibodies. Nuclei were stained with DAPI. Both vCyclin and HIF1α showed a clear distribution to the nuclear, cytosolic, or nucleo-cytosolic compartments. (**C**) Immuno-precipitation of vCyclin with HIF1α in HEK293T cells. GFP-vCyclin transfected cells were lysed in radio-immunoprecipiation buffer and protein were pre-cleared with protein agarose A/G beads followed by overnight immuno-precipitation with GFP antibody. The immune complexes were collected with Protein agarose A/G bead slurry. The beads were washed with PBS and were resuspended in 2X SDS loading dye. One-third of the immuno-precipitated complex was run on 10% SDS PAGE against 5% input or IgG control sample and probed with HIF1α or GFP (for vCyclin) antibodies. (**D**) HIF1α immuno-

*Figure 3 continued on next page*

*Figure 3 continued*

precipitation using vCyclin antibodies in KSHV positive BC3, BCBL1, and JSC1 cells. KSHV-negative BJAB cells were used as a negative control. (E) Western blot analysis for nuclear, cytosolic, or nucleo-cytosolic localization of HIF1α in vCyclin expressing cells. The transfected cells were lysed to separate into nuclear or cytosolic fraction as described in the materials and methods section. The nuclear or cytosolic fractions were checked for localization of HIF1α. GAPDH and Histone H4 served as loading controls for cytosolic and nuclear fraction, respectively. Asterisk represents non-specific bands.

The online version of this article includes the following figure supplement(s) for figure 3:

**Figure supplement 1.** KSHV-encoded vCyclin shows gradual cytosolic translocation in transfected cells.

eventually were lost even under antibiotics selection (*Figure 4C*). Further we performed a temporal analysis on vCyclin expression to determine its degradation dynamics. We transfected GFP-tagged vCyclin in HEK293T cells and checked its expression at 24 hr-time point intervals. As expected, the levels of GFP (tagged to vCyclin) were continuously decreased with its almost complete elimination at the seven day time point (*Figure 4—figure supplement 1*). The cytosolic translocation of expressed vCyclin, its elimination over time in cells, and its interaction with HIF1α led us to hypothesize that vCyclin may mediate degradation of HIF1α through the lysosomal pathway. Therefore, we treated cells expressing vCyclin with a lysosomal inhibitor (Chloroquine) with increasing concentrations. We also treated these cells with the proteasomal inhibitor (MG132) to rule out proteasomal degradation (*Hayashi et al., 1992*). The results clearly showed that treatment of cells expressing vCyclin with the proteasomal inhibitor MG132 failed to protect degradation of HIF1α. However, treatment of these cells with the lysosomal inhibitor Chloroquine protected HIF1α from vCyclin-mediated degradation (*Figure 4D*). Although, we observed a reduced expression of transfected vCyclin, as well as a higher level of HIF1α in the intermediate concentration of proteasomal inhibitor-treated cells, the former could be due to degradation of vCyclin itself in the lysosomal compartment, or it may be due to the cytotoxic/inhibitory effects of MG132. The results were also similar in the Saos-2 cell line (*Figure 4—figure supplement 1*), which further strengthens the finding that KSHV-encoded vCyclin is involved in degradation of HIF1α through the lysosomal pathway. To further corroborate these findings, we investigated the subcellular localization of HIF1α in cells expressing Myc-tagged KSHV-encoded vCyclin, especially with respect to its lysosomal localization. Interestingly, we observed that HIF1α localized to the nucleus as well as lysosomal compartments, as seen in the representative image of HIF1α in vCyclin expressing cells (*Figure 4E*). Nevertheless, vCyclin localization was also distributed to the nuclear as well as lysosomal compartment. This suggested a clear role of vCyclin in mediating nuclear to lysosomal translocation of HIF1α. To understand the relevance of vCyclin-mediated HIF1α attenuation, we further investigated the details of vCyclin expression under hypoxic conditions. We recently observed that KSHV-encoded vCyclin transcripts were upregulated in KSHV-positive BC3 cells grown under hypoxic conditions. We further corroborated these findings by using two KSHV-positive cells lines, BCBL1 and JSC1 (*Singh et al., 2018*). We treated these cells with the chemical mimetic of hypoxia, CoCl$_2$ for 8 hr to prevent VHL ubiquitin-ligase mediated degradation in normoxia, and stabilization of HIF1α in those cells (*Yuan et al., 2003*). HIF1α levels were monitored by western blot using HIF1α specific antibodies (*Figure 4—figure supplement 2*). The real-time PCR for vCyclin transcripts in these cells confirmed that hypoxia positively upregulated expression of KSHV-encoded vCyclin (*Figure 4—figure supplement 2*).

We further investigated the detailed mechanism by which HIF1α upregulated vCyclin expression and its negative feedback on HIF1α regulation. The promoter region of vCyclin was therefore analyzed for the presence of functional hypoxia responsive elements (HREs). A schematic for the KSHV latent gene locus (LANA, vCyclin and vFLIP) is shown (*Figure 4—figure supplement 2*). As the HREs within LTc or LTi promoter region are already studied and these HREs are common for the three latent genes (LANA, vCyclin, and vFLIP) (*Veeranna et al., 2012*), we focused on the HREs mainly in the promoter region downstream to them, and specific for ORF71 and ORF72 (LTd) (*Veeranna et al., 2012*).

The in-silico analysis of the vCyclin promoter (Ltd promoter) revealed the presence of four distinct HREs (*Figure 4—figure supplement 2*). To determine if these HREs are directly under the transcriptional regulation of HIF1α, we cloned these HREs into the luciferase reporter plasmid vector. Co-transfection of HIF1α with the promoter constructs was performed to determine which if any of the

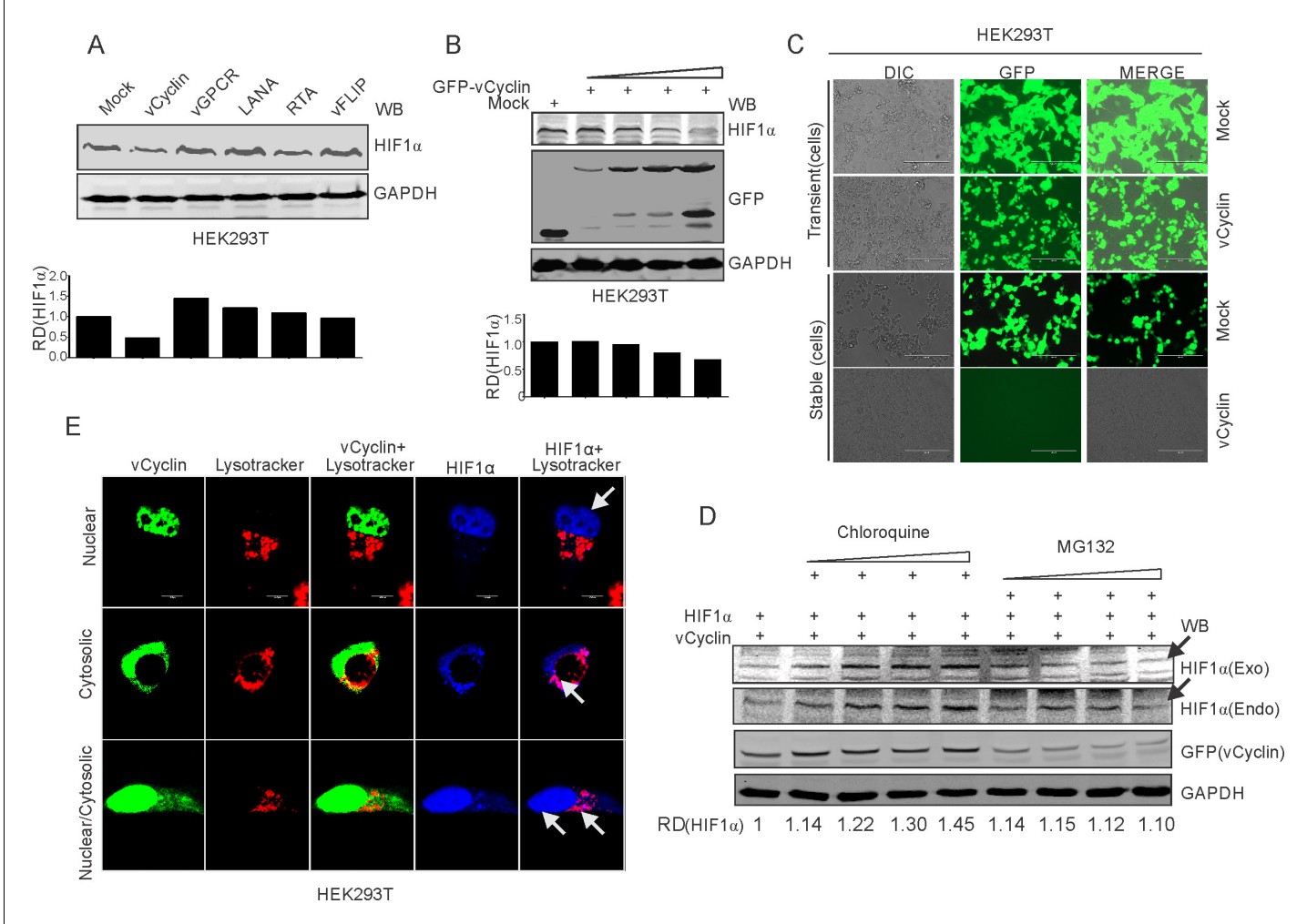

**Figure 4.** KSHV-encoded vCyclin degrades HIF1α in a dose-dependent manner through the lysosomal pathway. (**A**) Mock or KSHV-encoded vCyclin, LANA, RTA, vGPCR, and vFLIP were transfected into HEK293T cells followed by investigating HIF1α levels. Equal amounts of protein were used to probe HIF1α levels. GAPDH was used as the loading control (**B**) HIF1α, GFP-vCyclin and GAPDH western blots of cells transfected with gradually increasing amounts of GFP-vCyclin. GFP western blot represents GFP-vCyclin. GAPDH was used as the loading control. (**C**) Representative image for generation of mock or GFP-vCyclin stable cell lines (**D**) HIF1α western blot analysis of HEK293T cells transfected with GFP-vCyclin and treated with increasing amounts of lysosomal inhibitor (Chloroquine; 0, 6.25, 12.5, 25, and 50 µM) or proteasomal inhibitor (MG132; 1.25, 2.5, 5, and 10 µM). Twenty-four hr post transfection, culture media was removed and replaced with fresh medium containing either Choloroquine or MG132 as indicated. Arrow indicates HIF1α band. (**E**) Lysosomal translocation of HIF1α by KSHV-encoded vCyclin. GFP signals represents expressed vCyclin. Lysosomes were stained with Lysotracker red. HIF1α was probed with Alexa 350 (blue). Arrow indicates representative HIF1α localization.

The online version of this article includes the following source data and figure supplement(s) for figure 4:

**Figure supplement 1.** Dose-dependent lysosomal degradation of HIF1α by KSHV-encoded vCyclin in Saos-2 Cells.

**Figure supplement 2.** HIF1α-mediated upregulation of vCyclin in KSHV-positive BCBL1 and JSC1 cells.

**Figure supplement 2—source data 1.** vCyclin mediated degradation of HIF1α in naturally infected KSHV positive cell lines.

HREs within the Ltd promoter was predominantly responsible for vCyclin transactivation. The luciferase-based analysis suggested that the 1 st HRE is the most responsive in terms of elevated HIF1α levels (*Figure 4—figure supplement 2*). These results suggested a feedback regulation of HIF1α expression through KSHV-encoded vCyclin. These observations raised several questions such as, (1) whether or not HIF1α degradation happens in naturally infected cells, (2) whether knocking down vCyclin can rescue HIF1α degradation in KSHV-positive cells, and (3) whether vCyclin mediated lysosomal degradation is specific. To address these questions, we grew BC3-shControl and BC3-shvCyclin cells in normoxic or hypoxic conditions in either presence or absence of the lysosomal inhibitor, chloroquine. The results clearly suggested that lysosomal degradation of HIF1α do happen in

naturally infected cells. Furthermore, knock down of vCyclin allowed for higher expression of HIF1α (*Figure 4—figure supplement 2*).

## HIF1α enrichment on KSHV genome in hypoxic conditions

In hypoxia, HIF-1α is known to bind with Cdc6, a crucial protein required for loading cellular helicases (MCMs) onto DNA to initiate DNA replication (*Hubbi et al., 2013*). This also explains HIF1α-mediated repression of DNA replication in hypoxia. We hypothesized that the presence of KSHV may abrogate hypoxia-mediated negative regulation of DNA replication through vCyclin-dependent HIF1α degradation. To explore this, we investigated HIF1α enrichment on the KSHV genome using Chromatin Immunoprecipitation (ChIP) sequencing. As the degradation of HIF1α through expression of vCyclin did not represent a physiological condition for KSHV-infected cells growing under hypoxic conditions, we infected PBMCs with KSHV followed by growing mock or infected cells in normoxic or hypoxic conditions. KSHV infection was monitored by LANA immuno-fluorescence and was confirmed by LANA western blot (*Figure 5A and B*). We also included naturally infected KSHV positive BC3 cells for ChIP sequencing (*Figure 5C*). ChIP was performed using HIF1α antibody followed by

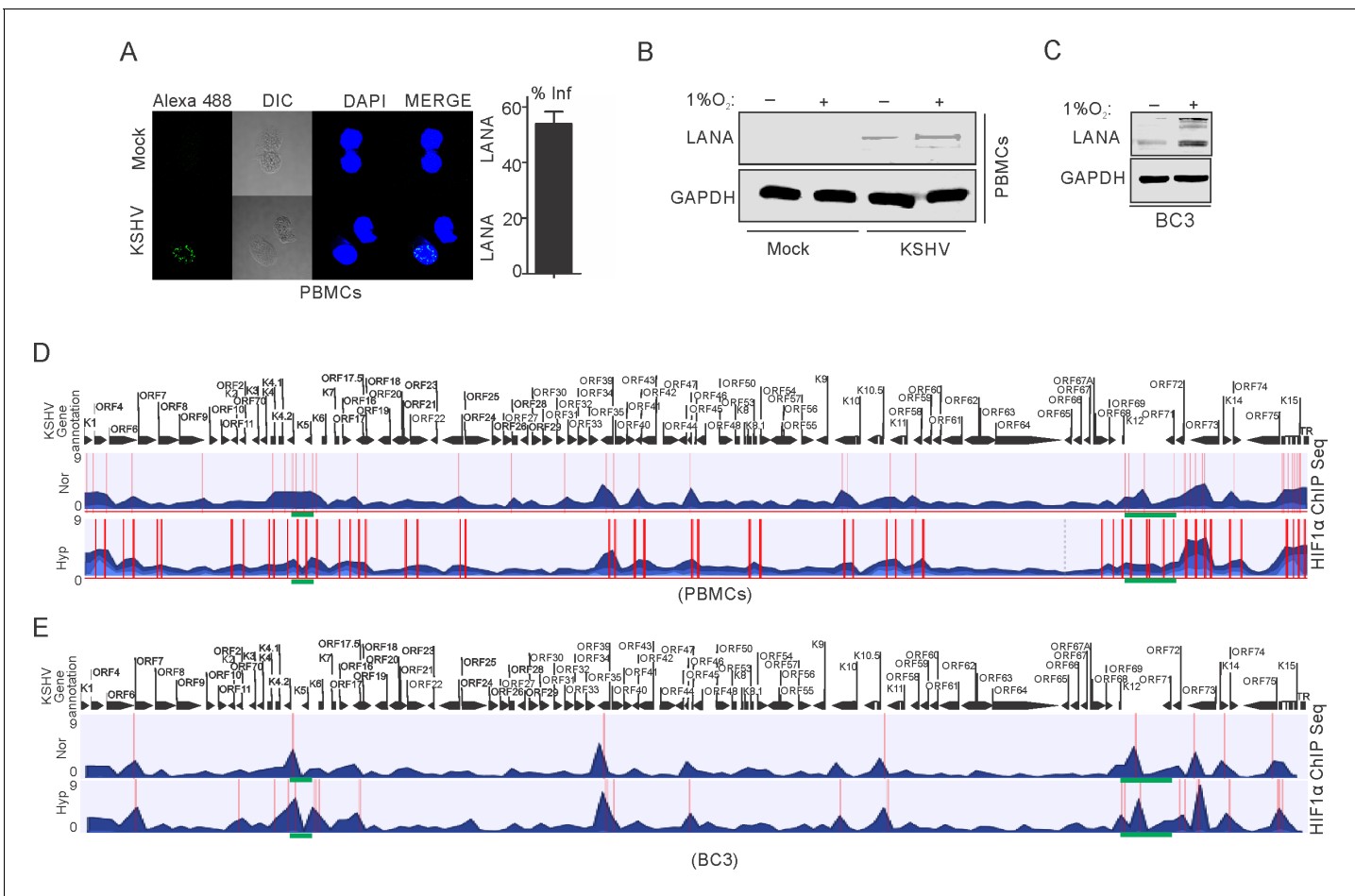

**Figure 5.** ChIP sequencing for HIF1α enrichment on KSHV genome in infected PBMCs or BC3 cells grown under normoxic or hypoxic conditions. (**A**) KSHV infection of PBMCs. BC3 cells were used to reactivate KSHV by TPA/BA treatment. Purified KSHV at a multiplicity of infection equal to 10 was used for infecting PBMCs in the presence of 20 μg/ml Polybrene for 4 hr followed by changing infection medium to fresh medium without Polybrene and grown for another 24 hr. Cells were divided into halves and grown either under normoxic or hypoxic conditions for 24 hr. A small fraction of cells were taken out to check the efficiency of infection by LANA immunofluorescence. The bar diagram represents the infection rate. (**B**) Western blot analysis of LANA in mock or KSHV-infected cells grown under normoxic or hypoxic conditions. GAPDH served as the loading control. (**C**) Western blot analysis of LANA in KSHV-positive BC3 cells grown under normoxic or hypoxic conditions. (**D and E**) Alignment of ChIP sequencing reads with KSHV genome in infected PBMCs and KSHV-positive BC3 cells grown under normoxic or hypoxic conditions. The ChIP sequencing data was analyzed using CLCbio software (Qiagen). The highlighted sections represent regions with significant differential enrichment.

next generation ChIP-sequencing. The ChIP-sequencing results for enrichment of HIF1α on the KSHV genome showed that even in the presence of KSHV, either in infected PBMCs or BC3, there is an almost fourfold higher enrichment of HIF1α on the KSHV genome in hypoxia (*Figure 5D and E*). Interestingly, the HIF1α-enriched regions were mostly located at regions distinct from the origin of lytic replication (located between K4.2 and K5, and between K12 and ORF71) (*Lin et al., 2003*; *Figure 5D,E* and *Supplementary file 2–5*). This suggests that during the period examined for hypoxic induction, there was likely no suppression of DNA replication through direct binding of HIF1α to the OriLyt region. Moreover, high enrichment of HIF1α on the KSHV genome can be an indicator of enhanced transcription of KSHV-encoded lytic genes required for KSHV reactivation (*Figure 5D and E*). Further, details of HIF1α enriched regions on the KSHV genome in normoxic or hypoxic conditions are provided in *Supplementary file 2–5*.

## Knock-down of vCyclin in KSHV positive cells attenuated KSHV-mediated bypass of hypoxia-repressed DNA replication and proliferation

We investigated the effects of knocking down KSHV-encoded vCyclin on DNA replication as well as proliferation of KSHV positive cells grown under hypoxic conditions. We generated lentivirus mediated vCyclin knock-down in KSHV positive BC3 cells (*Figure 6A*). The knock down of vCyclin was confirmed at the transcript level by real-time PCR using cDNA from ShControl or ShvCyclin transduced BC3 cells. A fourfold or greater downregulation of vCyclin was seen at the transcript level (*Figure 6B*). The BC3 ShControl or ShvCyclin cells were grown under normoxic conditions or in 1% $O_2$ to induce hypoxic conditions, and DNA replication and cell growth were examined. The effects of hypoxia-mediated repression of replication in vCyclin knockdown, KSHV-positive BC3 cells was investigated by copy number calculation of KSHV genomes in these cells grown under normoxia or hypoxia. As compared to BC3-ShControl, BC3-ShvCyclin cells showed an adverse effect on KSHV replication as evident from more than a 10-fold less KSHV copy number in hypoxia (*Figure 6C*). The results demonstrated the compromised replication potential of KSHV positive cells when vCyclin was knocked-down. Interestingly, knock-down of vCyclin adversely affected replication even under normoxic conditions (*Figure 6C*). We further validated these results at the level of single molecules of KSHV genome using single molecule analysis of replicated DNA (SMARD) (*Figure 6D and E*). In brief, BC3-ShControl, BC3-ShvCyclin cells were grown under hypoxic conditions followed by pulsing of these cells by nucleotide analogs to study their incorporation into replicating KSHV genomes. The linearized KSHV genomes on the slides were visualized with three KSHV-specific probes (KSHV co-ordinates: 26937-33194(6 kb); KSHV co-ordinates: 36883-47193(10 kb), and KSHV co-ordinates: 85820-100784(15 kb)) (*Verma et al., 2011*). A KSHV molecule with incorporated nucleotide analogs represented a genome that was replicated. The analysis of SMARD clearly demonstrated that knock-down of vCyclin suppressed KSHV replication by more than 10-fold in hypoxia (*Figure 6E and F*). To further strengthen the results and provide more specificity, we checked expression profile of vCyclin in naturally infected BC3 cells grown under hypoxic conditions and investigated whether it mediate HIF1α degradation and if it correlated with decreased levels of HIF1α. The results clearly showed that increased expression of vCyclin directly correlated with HIF1α degradation (*Figure 6—figure supplement 1*).

We then investigated the potential of vCyclin knock down in KSHV-positive cells on anchorage-dependent and -independent growth. ShControl or ShvCyclin HEK293T-BAC16-KSHV cells were used to generate anchorage-dependent colonies, while ShControl or ShvCyclin BC3 cells were used to generate anchorage-independent colonies in soft agar. Both types of colonies were challenged to grow in hypoxia for 72 hr followed by staining and visualization by trypan blue staining. The results showed that ShvCyclin cells were highly compromised for both anchorage dependent, and independent growth as evident from the size and number of colonies (*Figure 6G*). These results clearly indicated a role for vCyclin in balancing HIF1α levels in infected cells to allow proper replication and growth under hypoxic conditions.

## Discussion

Hypoxia represents a detrimental stress to aerobic cells and is an established physiological character of cancer as well as oncogenic virus infected cells (*Seyfried and Shelton, 2010*; *Noch and Khalili,*

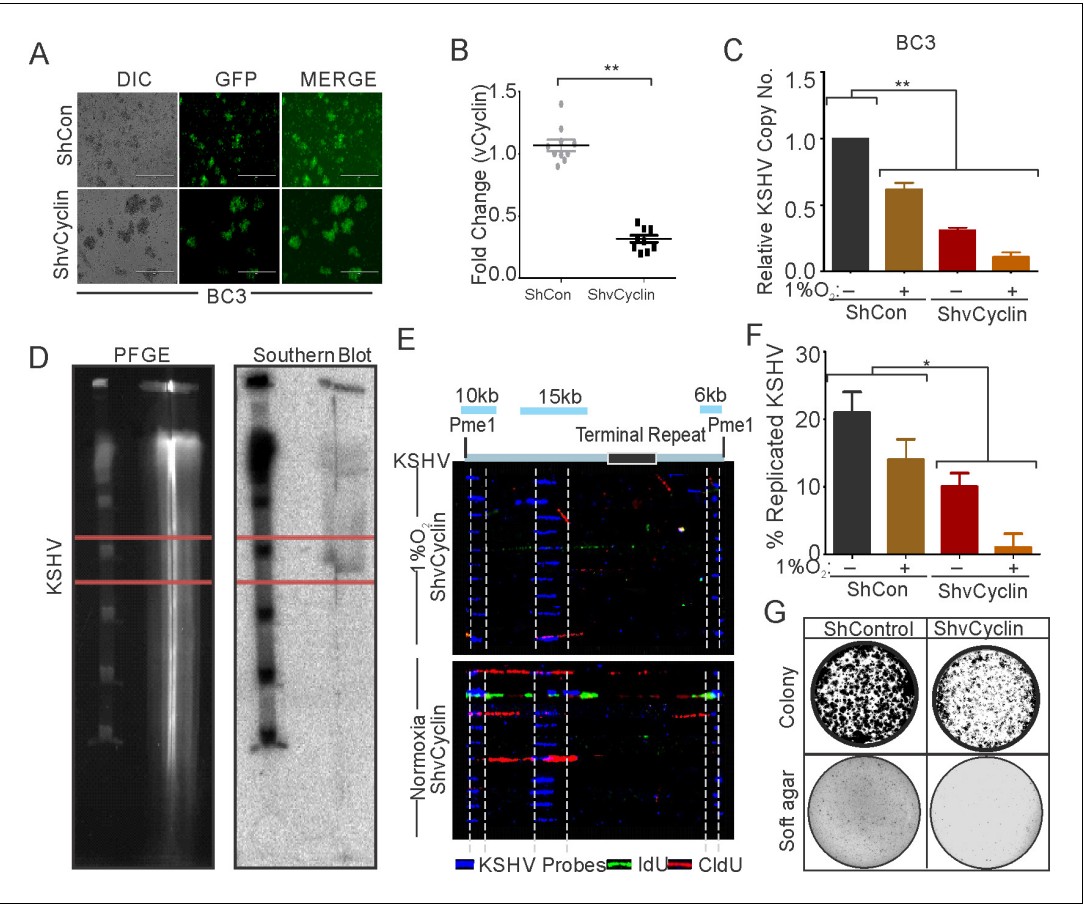

**Figure 6.** vCyclin knock down in KSHV-positive cells shows compromised growth potential in hypoxic conditions. (A) Generation of vCyclin knock down stable cells in KSHV-positive BC3 cells. BC3 cells were transduced with ShvCyclin lentiviruses and selected in puromycin for 3 weeks. GFP-positive cells represent successfully transduced stable cells. (B) Real-time PCR for vCyclin in BC3 shControl and ShvCyclin stable cells (n=9; asterisk represents statistically significant difference). (C) Relative copy number calculation of KSHV in BC3 shControl and ShvCyclin stable cells grown under normoxic or hypoxic conditions. The experiments were performed in triplicate. The error bar represents standard error from the mean. A p-value of <0.05, was taken into consideration for statistical significance, (** p<0.001). (D) Representative image for pulse field gel electrophoresis and Southern blot analysis for nucleotide analog pulsed cells after proteinase K and Pme1 digestion. (E) Representative image for single molecule analysis of replicated DNA of BC3 ShControl or BC3-ShvCyclin cells grown under hypoxic conditions. Blue fluorescence represents KSHV specific probes; red represents incorporation of IdU, while green represents CldU incorporation. (F) Bar diagram for quantitation of replicated KSHV molecules in BC3 ShControl or BC3-ShvCyclin cells grown under normoxic or hypoxic conditions. (G) Colony focusing and soft agar assay to investigate in vitro anchorage-dependent/independent grown potential of shControl and ShvCyclin of KSHV-positive stable cells grown under hypoxic conditions.

The online version of this article includes the following source data and figure supplement(s) for figure 6:

**Source data 1.** vCyclin is essential for hypoxic reactivation of KSHV.

**Figure supplement 1.** Degradation of HIF1α in naturally KSHV-infected cells grown under normoxic or hypoxic conditions at various time points.

2012). The hypoxic cells, not only need large scale reprogramming of cellular pathways for survival, but also poses a major challenge for patients undergoing chemotherapy and radiation therapy (*Raz et al., 2014*). The well-known factors responsible for reprogramming of cellular pathways under hypoxic conditions include the hypoxia inducible factors (HIFs) which possess broad transcriptional capacity for cells grown in hypoxia (*Majmundar et al., 2010*). HIF1α is considered a rate limiting protein based on its stability, specifically under hypoxic conditions (*Majmundar et al., 2010*). When stabilized, HIF1α transactivates genes involved in, but not limited to those associated with glucose

transport (GLUTs), glycolysis (HK1, HK2, ENO1, GPI, and LDHA), angiogenesis (VEGF, ENG, and LRP1), growth and survival (Cyclin G2, TGF-b, EPO, and NOS), and invasion and metastasis (KRTs, MMP2, UPAR, AMF, and PAI1) (*Dengler et al., 2014*; *Majmundar et al., 2010*; *Ortiz-Barahona et al., 2010*). Based on these broad roles of HIF1α in growth and survival, the protein is often considered an oncogenic protein (*Bárdos and Ashcroft, 2004*). Under favorable normoxic conditions, the protein levels of HIF1α is tightly regulated through its proteasomal degradation by the tumor suppressor protein VHL or lysosomal degradation mediated by cyclin-dependent kinase 2 (CDK2) (*Hubbi et al., 2014*; *Tanimoto, 2000*; *Pugh and Ratcliffe, 2003*).

KSHV, like other oncogenic viruses, utilizes multiple mechanisms to up-regulate and stabilize HIF1α (*Cuninghame et al., 2014*; *Zinkernagel et al., 2007*; *Semenza, 2012*; *Carroll et al., 2006*; *Catrina et al., 2006*). Elevated HIF1α in KSHV infected cells is known to modulate expression of key oncogenes encoded by KSHV in an autocrine manner, in addition to transactivation of cellular genes responsible for survival, growth, proliferation, angiogenesis, and metabolic reprogramming (*Ma et al., 2015*; *Jham and Montaner, 2010*; *Jham et al., 2011*; *Semenza, 2010*; *Delgado et al., 2010*). KSHV-encoded vGPCR effects on the p38/MAPK pathway, and EC5S-mediated ubiquitination of the tumor suppressor VHL by KSHV-encoded LANA are extensively studied pathways which shows upregulated expression of HIF1α in KSHV-infected cells (*Cai et al., 2006b*; *Sodhi et al., 2000*). HIF1α also upregulates vGPCR in a cyclic manner to maintain continued elevated levels of HIF1α, and targeting HIF1α in KSHV infected cells has been proposed as a potential therapeutic strategy against KSHV-mediated oncogenesis (*Singh et al., 2018*; *Shrestha et al., 2017*). It is noteworthy that hypoxia or elevated HIF1α levels induces arrest of cell cycle and DNA replication to minimize energy utilization as well as the macromolecular demands on cells, and this arrest is crucial for survival until favorable conditions return. Nevertheless, HIF1α-mediated inhibition of cell cycle progression and DNA replication depends on both transcriptional, as well as non-transcriptional activity of HIF1α (*Gardner et al., 2001*; *Hammond et al., 2002*; *Goda et al., 2003*; *Hubbi et al., 2013*; *Box and Demetrick, 2004*; *Green et al., 2001*; *Martin et al., 2012*).

Paradoxically, in contrast to healthy cells with arrested cell cycle and DNA replication, KSHV-infected cells not only divide and replicate in hypoxia, but is induced to productive (lytic) replication (*Davis et al., 2001*; *Cai et al., 2006a*; *Gardner et al., 2001*; *Goda et al., 2003*). Interestingly, how KSHV bypasses hypoxia-mediated repression of cell cycle and DNA replication to promote its hypoxic reactivation has never been studied adequately. This is partly due to the fact that KSHV based research have been hindered by a lack of isogenic KSHV-negative control cell lines. Infection based studies also include constraints with variable expression of KSHV-encoded genes, as well as the initial epigenetic reprogramming of the KSHV genome upon entry into infected cells. In the present study, we used KSHV negative BJAB cells matched with its counterpart BJAB-KSHV cells (BJAB cells stably transfected with BAC-KSHV) (*Chen and Lagunoff, 2005*). These cell lines were characterized for their isogenic background through short tandem repeats (STR) profiling (*Singh et al., 2018*). Additionally, BJAB-KSHV cells were checked for typical characteristics of latently infected KSHV-positive cells and a similar passage number of cells were used in this study to rule out any further constraints (*Singh et al., 2018*). Replication rate analysis of these two cell lines grown under hypoxic conditions clearly demonstrated that the presence of KSHV helped in bypassing hypoxia-mediated replication arrest despite having a higher level of HIF1α in normoxia and hypoxia. The higher levels of HIF1α appeared to be aligned with its transcriptional activity where these cells have significantly higher levels of HIF1α targeted downstream transcripts such as PDK1, LDHA and P4HA. Interestingly, the introduction of hypoxia to these cells through growth in low oxygen resulted in significantly less upregulation of these HIF1α targets in BJAB-KSHV cells compared to matched KSHV-negative BJAB cells. This result suggested that BJAB-KSHV cells were adjusted for growth with higher levels of HIF1α and show the same growth pattern as KSHV-negative BJAB cells in normoxic conditions. However, the presence of KSHV negatively regulated the transcriptional activity of HIF1α in hypoxic conditions to provide an advantage for cell proliferation compared to KSHV-negative BJAB cells. A similar result was seen in PBMCs infected with mock or rKSHV and further confirms the role of the virus in DNA replication of infected cells grown under hypoxic conditions. In addition, it provides new insights into a role for KSHV latent antigens, which was also expressed upon infection as well as under hypoxic conditions, with a major role in this transition.

The KSHV-encoded LANA, RTA, vGPCR, vCyclin, and vFLIP are well-established antigens expressed during latency, initial infection or under hypoxia induction. Expression of these antigens

individually when compared to mock revealed that vCyclin can efficiently downregulate the transcriptional activity of HIF1α. Additionally, expression of GFP-tagged vCyclin in cells showed a variable localization within either nuclear, cytosolic or both nuclear and cytosolic compartments. Further, long-term expression of vCyclin was unsuccessful after selection in culture. These results provide new insights into a role for collaboration of vCyclin with HIF1α in mediating its cytosolic translocation, and the degradation of both vCyclin and HIF1α within the lysosomal compartment. Immunoprecipitation of HIF1α with vCyclin, the co-localization of both vCyclin and HIF1α to the lysosomal compartment in vCyclin expressing cells, and their degradation as well as their protection from degradation in cells treated with Chloroquine, the lysosomal inhibitor, clearly demonstrated a role for vCyclin in regulation of lysosomal degradation of HIF1α in KSHV-positive cells in hypoxia. It is important to note that, in KSHV-positive cells, HIF1α itself upregulated expression of vCyclin through HREs transactivation forming a negative feedback control (*Figure 7*). Adding these finding with already known facts such as CDK2 mediation of HIF1α degradation through lysosomal pathway (*Hubbi et al., 2014*), and that HIF1α can act as an inhibitor of DNA replication (*Hubbi et al., 2013*), we can expect to target the phenomenon of lysosomal degradation of HIF1α using Chloroquine to investigate the possible strategy for therapeutic treatment against KSHV-mediated tumorigenesis.

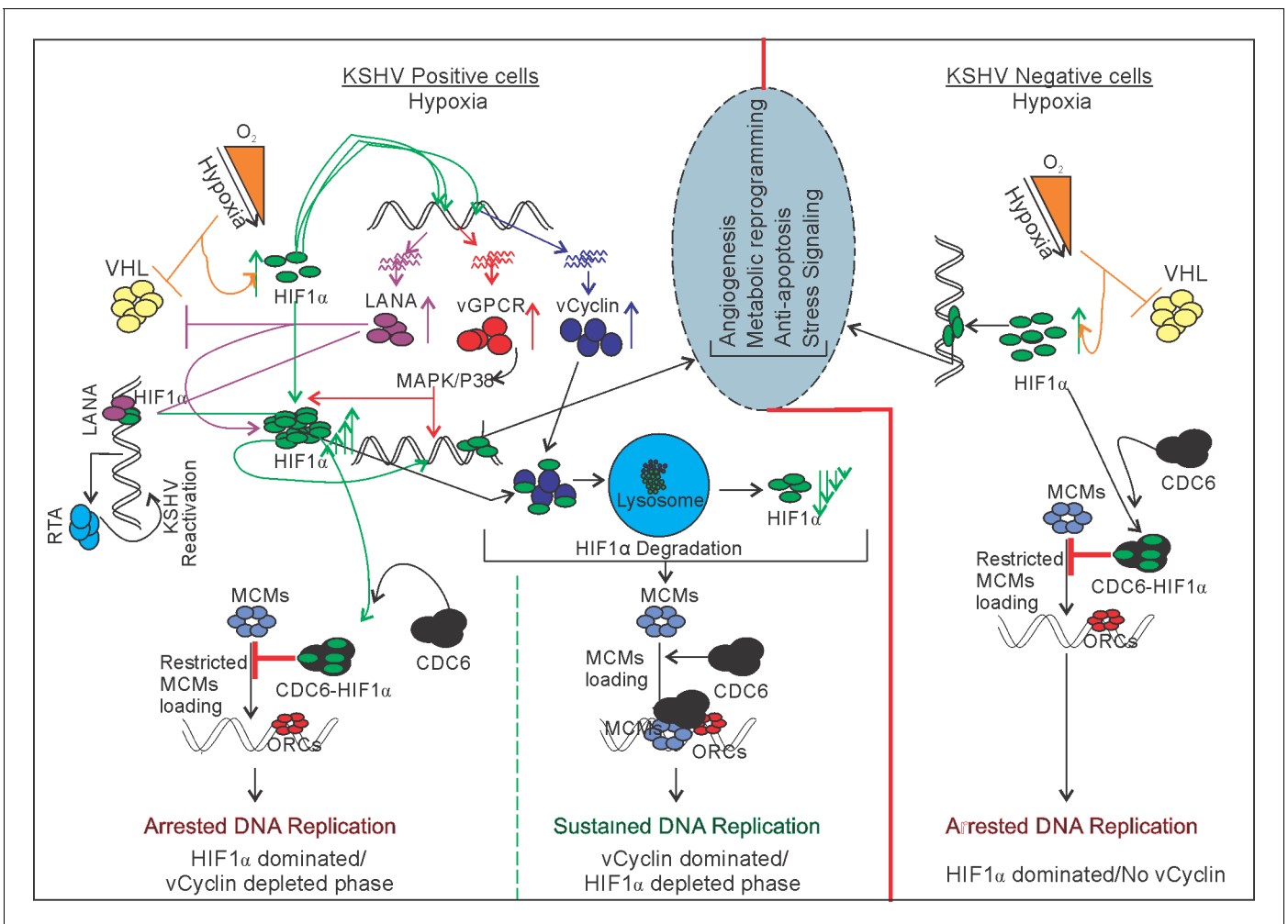

**Figure 7.** Schematic for KSHV-mediated attenuation of HIF1α-mediated repression of DNA replication in hypoxic conditions. A KSHV uninfected cell shows upregulated expression of HIF1α which restricts DNA replication in hypoxic conditions. In KSHV-infected cells, KSHV-encoded LANA and vGPCR promotes upregulated expression of HIF1α through inhibition of its proteosomal degradation or by working on MAPK/P38 pathway, respectively. KSHV-encoded vCyclin, which is also expressed in response to hypoxia, interacts and regulates HIF1α transcriptional activity, as well as degrades it through the lysosomal pathway to balance HIF1α-mediated restriction of DNA replication in hypoxia.

ChIP-Sequencing results of KSHV-infected cells grown under normoxic or hypoxic conditions to define the transcriptional activity or repression of replication due to HIF1α in these cells under physiologically relevant conditions showed significantly high reads across the KSHV genome in hypoxia, especially near the latency locus. Further analysis of the ChIP-Seq results provided ample information about the regulation of HIF1α transcriptional activity on the KSHV genome. Further, it provides an explanation for how the KSHV lytic origin of replication was derepressed by HIF1α-mediated blockage of replication (Green highlighted area at the bottom of the histogram) (*Figure 5*). Overall, the present studies clearly demonstrated that KSHV-encoded vCyclin attenuates HIF1α signaling in a negative feedback loop to create a favorable microenvironment where DNA replication is permitted. The work also explores a new area of investigation as to the role of another oncogenic viral encoded antigen in driving DNA replication in hypoxia. It also extends our knowledge as to the stringent regulation of HIF1α required for KSHV genome replication and cell proliferation by three KSHV-encoded antigens and explores their complementary roles for viral genome persistence and survival of the infected cells in the hypoxic microenvironment. Other challenging questions behind hypoxic reactivation of KSHV remain unexplained and unexplored. For example, hypoxia also induces suppression in transcription, translation, as well as effects on epigenetic regulation and promotion of protein degradation. We recently showed that KSHV-encoded-LANA protects replication associated proteins from hypoxia mediated degradation (*Singh et al., 2019*). This study further adds to our understanding of the mechanism of hypoxic reactivation of KSHV. Nevertheless, much more studies are needed to provide a more comprehensive scenario that surrounds hypoxic reactivation of KSHV. This includes investigation of the maintenance of transcription of essential genes required for replication, as well as epigenetic reprogramming which occurs in hypoxia during KSHV replication. Additionally, development of new technologies and strategies to investigate these finding in vivo will be a milestone achievement in assessing the role of KSHV in regulating HIF1α for continuous replication under hypoxic conditions of many types of cancers. Although, this current study provides insights into the replication strategy of KSHV infected cells grown under hypoxic conditions, the replication strategy for uninfected cells remains an open question. However, we can speculate about the existence of mechanisms that stabilize Cyclins and CDKs under hypoxic conditions in non-infected cells, potentially regulating HIF1α levels in non-infected cells. Hence, this study further opens a new dimension to the role of viruses in replication and proliferation of other cancer types where HIF1α expression is induced in infected cells.

# Materials and methods

## Key resources table

| Reagent type (species) or resource | Designation | Source or reference | Identifiers | Additional information |
|---|---|---|---|---|
| Cell line (*Homo sapiens*) | BJAB | Harvard Medical School, Boston, MA | PMID:2157887 | Gift from Dr. Elliott Kieff (Harvard Medical School, Boston, MA) |
| Cell line (*Homo sapiens*) | BJAB-KSHV | University of Washington, Seattle, Washington | PMID:16254372 | Gift from Dr. Michael Lagunoff (University of Washington, Seattle, Washington) |
| Cell line (*Homo sapiens*) | BC3 | American Type Culture Collection | CRL:2277 | https://www.atcc.org/products/crl-2277 |
| Cell line (*Homo sapiens*) | BCBL1 | American Type Culture Collection | PMID:8612236 | Gift from Dr. Don Ganem (University of California School of Medicine, San Francisco, CA) |

*Continued on next page*

*Continued*

| Reagent type (species) or resource | Designation | Source or reference | Identifiers | Additional information |
|---|---|---|---|---|
| Cell line (*Homo sapiens*) | JSC1 | Johns Hopkins University School of Medicine, Baltimore, MD | CRL-2769 | Gift from Dr. Richard F. Ambinder (Johns Hopkins University School of Medicine, Baltimore, MD) |
| Cell line (*Homo sapiens*) | HEK293T | Brigham and Woman's Hospital, Boston, MA | CRL:3216 | Gift from Dr. John Aster Brigham and Womens Hospital, Boston, MA |
| Cell line (*Homo sapiens*) | Saos-2 | Brigham and Woman's Hospital, Boston, MA | HTB:85 | Gift from Dr. John Aster Brigham and Womens Hospital, Boston, MA |
| Antibody | Anti-HIF1α (Mouse monoclonal) | Novus Biologicals | Cat No. NB 100–105 | WB: 1:250 IF: 1:50 |
| Antibody | Anti-GAPDH (Mouse monoclonal) | Santa Cruz Biotechnology | Cat No. SC-32233 | WB: 1:2000 |
| Antibody | Anti-myc (Mouse monoclonal) | Hybridoma | Purified antibody | WB: 1:250 |
| Antibody | Anti-GFP (Mouse monoclonal) | Santa Cruz Biotechnology | Cat No. Sc-9996 | WB: 1:2000 |
| Antibody | Anti-vCyclin (Rat monoclonal) | Abcam | Cat No. ab12208 | WB: 1:250 IF: 1:50 |
| Antibody | Anti-LANA (Mouse monoclonal) | Hybridoma | Purified antibody | WB: 1:250 IF: 1:100 |
| Antibody | Anti-IdU (Mouse monoclonal) | BD Fisher Scientific | Cat No. BDB347580 | IF: 1:20 |
| Antibody | Anti-CldU (Rat monoclonal) | Accurate Chemical | Cat No. OBT0030S | IF: 1:20 |
| Antibody | Alexa flour 488 (Goat polyclonal) | ThermoFisher | Cat No. 11006 | IF: 1:20 |
| Antibody | Alexa flour 568 (Goat polyclonal) | ThermoFisher | Cat No. 11004 | IF: 1:20 |
| Antibody | Anti-Histone 4 (Rabbit polyclonal) | Merck | Cat No. 07–108 | WB: 1:1000 |
| Chemical compound, drug | YOYO-1 | ThermoFisher | Cat No. Y3601 | |
| Chemical compound, drug | Biotinylated anti-avidin | Vector Laboratory | Cat No. BA-0300 | |
| Chemical compound, drug | Neutr-avidin alexa 350 | ThermoFisher | Cat No. A11236 | |
| Chemical compound, drug | IdU | Sigma-Aldrich | Cat No. I7125-5G | |
| Chemical compound, drug | CldU | Sigma-Aldrich | Cat No. C6891 | |
| Chemical compound, drug | Transfection Reagent | Polyplus Transfection | Reference No. 114–07 | |
| Chemical compound, drug | MG132 | abcam | Cat No. ab141003 | |

*Continued on next page*

*Continued*

| Reagent type (species) or resource | Designation | Source or reference | Identifiers | Additional information |
|---|---|---|---|---|
| Chemical compound, drug | Chloroquine | Sigma-Aldrich | Cat No. C6628 | |
| Chemical compound, drug | DAPI | Sigma-Aldrich | Cat No. D9564 | |
| Chemical compound, drug | Lysotracker | ThermoFisher | Cat No. L12492 | |
| Software, algorithm | GraphPad Prism | GraphPad Prism (http://graphpad.com) | RRID:SCR_015807 | |
| Software, algorithm | ImageJ | ImageJ (http://imagej.nih.gov/ij/) | RRID:SCR_003070 | |
| Software, algorithm | CLC Bio | Qiagen (http://qiagen.com) | Cat. No. / ID: 832021 | |

## Cell lines, plasmid vectors, transfection, and lentivirus transduction

Cell lines were obtained from ATCC or received as a gift. The cells lines were authenticated by STR profiling. Regular testing for mycoplasma contamination was performed to confirm contamination free culture. The details of cell lines and chemical reagents is provided in key resources table. In brief, BJAB, BJAB-KSHV, BC3, BCBL1, JSC1, and PBMCs were maintained in RPMI medium supplemented with 10% bovine growth serum (BGS) and antibiotics. HEK293T cells were maintained in DMEM medium with 5% BGS and antibiotics. Constructs for Myc-tagged LANA and Myc-tagged RTA constructs were described earlier (*Cotter and Robertson, 1999*). KSHV-encoded vCyclin and vFLIP were amplified by PCR, and ligated to the EcoR1/Xho1 restriction site of the pA3M vector. KSHV-encoded vGPCR was amplified by PCR from pCEFL-vGPCR plasmid vector (a gift from Enrique A Mesri, University of Miami) and sub-cloned into the EcoR1/Xho1 restriction site of pA3M plasmid vector. GFP-tagged-vCyclin and constructs were generated by PCR amplification and cloned into the pLVX-Ac-GFP vector. Transfections were carried out using jetPRIME reagent (Polyplus Transfection Inc, New York, NY) or through Calcium-phosphate based methods. ShRNA targeting KSHV-encoded vCyclin was generated using the pGIPZ vector. For production of lentivirus, pGIPZ clones along with packaging and helper plasmids were transfected into HEK293T cells as described earlier (*Singh et al., 2018*). Lentivirus transduction was performed in a total volume of 1 ml in the presence of 20 μg/ml Polybrene. Unless otherwise stated, all cultures were incubated at 37°C in a humidified environment with 5% $CO_2$. For hypoxic induction, cells were grown in the presence of 1% $O_2$ in a humidified chamber. The LTd promoter and deletion constructs for various hypoxia responsive elements (HREs) namely, HRE1, HRE2, HRE3, and HRE4 were cloned into the luciferase reporter plasmid pGL3-basic vector (Promega Corporation, Madison, WI).

## KSHV virion purification and infection of PBMC

KSHV-positive BC3 cells were used for production of virus through reactivation and purification of KSHV for infections. Briefly, cells were grown in culture medium containing TPA and butyric acid at a final concentration of 20 ng/ml and 3 mM, respectively for 4–5 days. Cells and the medium were collected and centrifuged at 3,000 rpm for 30 min to collect supernatant. The cell pellet was resuspended in 1X PBS and the cells were ruptured by freeze thaw, followed by centrifugation at 3000 rpm for 30 min. The supernatant were collected and pooled followed by filtration through 0.45 μm filter. The viruses were concentrated by ultracentrifugation at 23,500 rpm at 4°C for 2 hr. The KSHV genomic region (36883–47193) (*Verma et al., 2011*) cloned in pBS-puro plasmid was used to generate a standard curve for copy number calculation of the purified virus. PBMCs were infected with purified KSHV virions at a multiplicity of infection of 10 in a total volume of 1 ml in the presence of

20 µg/ml Polybrene. The rate of infection was monitored by immuno-fluorescence to detect LANA expression.

## Western blotting and immunoprecipitation

Primary antibody against Myc-tag was purified from 9E10 hybridoma (a gift from Dr. Richard Longnecker, Northwestern University). Antibodies for GFP and GAPDH were obtained from Santa Cruz Biotechnology (Dallas, TX). Histone H4 antibodies were procured from Millipore (Burlington, MA). Antibodies used against LANA was purified hybridoma (a gift from Dr. Ke Lan, Key State Virology Laboratory, Wuhan, China). HIF1$\alpha$ antibodies were obtained from Novus Biologicals (Centennial, CO). vCyclin antibodies were obtained from Abcam (Cambridge, MA). Cells were lysed with radio-immunoprecipitation (RIPA) buffer (50 mM Tris; pH 7.6, 150 mM NaCl, 2 mM EDTA, 1% Nonidet P-40) supplemented with protease inhibitors (Aprotinin [1 g/ml], Leupeptin [1 g/ml]and Pepstatin [1 g/ml] and phenylmethylsulfonyl fluoride (PMSF)[1 mM]). Equal amounts of protein were fractionated on SDS-PAGE followed by transfer to nitrocellulose membrane. Protein transfer was monitored through Ponceau staining and the membrane was washed with TBST followed by blocking in 5% skimmed milk. Primary antibody incubation was performed at 4°C overnight with gentle shaking. Infrared conjugated secondary antibodies were used to probe and capture protein levels using an Odyssey scanner (LiCor Inc Lincoln, NE). For immunoprecipitation, transfected cells were lysed in RIPA buffer and proteins were pre-cleared with protein A/G agarose beads. The pre-cleared lysates were incubated with the specific antibodies overnight at 4°C with gentle shaking. Protein A/G agarose beads slurry were used to collect the immune complexes. The beads were washed three times with 1X phosphate buffer saline followed by resuspension in equal volumes of 2X SDS loading dye. Approximately, 5% sample was used as input control.

## Fractionation of nuclear and cytoplasm proteins

Transfected cells were harvested by scraping on ice-cold PBS and collected by centrifugation at 1000 rpm for 5 min. The supernatants were discarded and the cell pellet were resuspended in hypotonic lysis buffer (20 mM HEPES-K$^+$ [pH 7.5], 10 mM KCl, 2 mM MgCl$_2$, 1 mM EDTA, 1 mM EGTA, 0.5M dithiothreitol) in the presence of protease inhibitors (1mMphenylmethylsulfonyl fluoride, aprotinin [1 g/ml], leupeptin [1 g/ml]and pepstatin [1 g/ml], 10 µg of phenanthroline/ml, 16 µg of benzamidine/ml) for 20 min. Samples were passed through 27-gauge needle 10 times before nuclei were pelleted by centrifugation at 3000 rpm for 5 min. The cytosolic supernatant was collected and centrifuged at 15,000 rpm for 15 min at 4°C to obtain the clear lysate. The nuclear pellet were resuspended in lysis buffer for a second time and passed through 25-gauge needle for 10 times followed by centrifugation at 3,000 rpm for 10 min. Nuclear pellet was resuspended in 1X TBS, 0.1% SDS. Genomic viscosity was removed by sonication of the samples using QSONICA sonicator (Newtown, CT).

## Quantitative real-time PCR

Total RNA from cells was extracted through standard phenol chloroform extraction using Trizol. RNA concentration and quality were measured using a biophotometer and on multimode spot reader using a Cytation 5 (BioTek Inc Winooski, VT). Total RNA from different samples were used to synthesize cDNA using Superscript II reverse transcription kit (Invitrogen, Inc, Carlsbad, CA). One µl of 10X dilution of cDNA was used per reaction for real-time quantitative PCR in a total volume of 10 µl using SYBR green reagent. A melting curve analysis was performed to verify the specificity of the amplified products. The relative fold change in expression were calculated by the delta delta threshold cycle method and each sample were measured in triplicates. The sequences of primers used for real-time PCR are given in *Supplementary file 1*.

## Immunofluorescence

Cells on a glass coverslip were fixed using 4% PFA for 20 min at room temperature followed by washing in 1X PBS. A combined blocking and permeabilization of cells were performed for 1 hr in 1X PBS containing 0.1% Triton X-100 with 5% serum from the species used to generate secondary antibody. Primary antibodies were diluted in 1X PBS/0.1% Triton X-100 and incubated overnight at 4°C followed by three times washing in 1X PBS. Secondary antibodies were diluted in 1X PBS at a

concentration suggested by manufacturer and incubated at room temperature for 1 hr followed by three times wash in 1X PBS. Nuclei were stained using DAPI at a final concentration of 1 µM for 20 min at room temperature followed by washing in 1X PBS. The slides were mounted using anti-fade reagent and sealed with colorless nail polish. The slides were kept at −20°C until use for capturing images with an Olympus confocal microscope (Olympus Corp., Tokyo, Japan).

## Chloroquine and MG132 treatment, and Lysotracker staining

The cytotoxic effect of Chloroquine and MG132 was determined based on previously published studies (*Rossi et al., 2007*; *Han et al., 2009*). Chloroquine was used at a concentration of 50 µM, and 5 µM was used as a maximum concentration of MG132 to treat cells. Twenty-four hr post-transfection, the culture medium was replaced with fresh medium containing either Chloroquine or MG132. For Chloroquine treatment, a wide range of concentration (0, 6.25, 12.5, 25, and 50 µM) was used for 24 hr. Similarly, for MG132, the range of concentration used was 0.625, 1.25, 2.5, and 5 µM and the treatment time was for 24 hr. Lysotracker staining was performed on live cells grown on cover slips at 75 nM final concentration of Lysotracker Deep red (ThermoFisher Scientific Inc, Grand Island, NY).

## Colony formation and soft agar assays

For colony formation and soft agar assays, KSHV positive BC3 cells were transduced with lentivirus coding either ShControl or ShvCyclin. The transduced cells were selected with puromycin for 3 weeks. 100% stably transduced (GFP positive) cells were used for colony formation and soft agar assay. Ten million HEK293T-KSHV cells were transfected with pGIPZ-shCon, shHIF1α or shvCyclin plasmids by electroporation and seeded in 10 cm dishes. Transfected cells were grown in DMEM containing puromycin at 0.5 µg/ml concentration. After selecting the cells for up to 2 weeks, selected cells were fixed with 4% formaldehyde and stained with 0.1% Crystal Violet solution (Sigma-Aldrich Corp., St. Louis, MO). The area of the colonies was calculated by using Image J software (Adobe Inc, San Jose, CA). The data shown here represents the average of three independent experiments.

## Luciferase reporter assays

Full-length LTd promoter or deletion construct for different HREs were cloned into pGL3 basic vector between Mlu1 and Xho1 restriction sites through PCR-based method. HEK293T cells were transfected with either full-length LTd promoter or deletion constructs of HREs. The cells were harvested at 48 hr post-transfection and subsequently washed once with PBS, followed by lysis with 200 µl of reporter lysis buffer (Promega, Inc Madison, WI). Forty µl of the total protein lysate was mixed with 25 µl of luciferase assay reagent. Luminescence was measured on a Cytationfive multimode reader (BioTek, Inc Winooski, VT). Relative luciferase activity was expressed as fold activation relative to that of the reporter construct alone. The results shown represent assays performed in triplicate. The sequences of primers used to clone full-length LTd promoter or various HREs deletion constructs are given in *Supplementary file 1*.

## Statistical analysis

All experiments were performed in replicates of three. The results were presented as means ± standard error of the mean (SEM). The data was considered statistically significant when the p-value was <0.05 using the Student's t-test.

## Single molecule analysis of replicated DNA (SMARD)

Single molecule analysis of replicated DNA of KSHV was performed as described earlier (*Verma et al., 2011*). In brief, cells pulsed with nucleotide analog (Chloro-Uridine, CldU, 10 µM and Iodo-Uridine, IdU, 10 µM) were mixed with InCert agarose to make plugs. Plugs were digested with Proteinase K followed by linearizing the KSHV genome by digesting with Pme1 restriction enzyme. Plugs were run on pulse field gel electrophoresis (PFGE). A portion of gels were used to perform Southern blot to locate and excise the gel regions enriched with KSHV genome. Gel slices were digested with agarose and DNA was stretched on silanized glass slides. The KSHV genome was

probed with KSHV-specific hybridization probes, and incorporation of CldU and IdU was visualized after probing with specific antibodies.

## ChIP sequencing

Cells were grown in either normoxia or 1% $O_2$ induced hypoxia. Cells were cross-linked by adding formaldehyde to a final concentration of 1% at room temperature. Crosslinking was stopped by adding glycine with shaking. Cells were collected by centrifugation and washed three times before resuspending in PBS. Cell pellets were resuspended in cell lysis buffer (5 mM PIPES pH8.0/85 mM KCl/0.5% NP-40) containing protease inhibitors (1 µg/ml Aprotinin, 1 µg/ml Leupeptin, 1µ/ml Pepstatin, and 1 mM PMSF). Cells were incubated on ice and homogenized with several strokes using a douncer. Nuclei were collected by centrifugation and lysed by adding nuclear lysis buffer (50 mM Tris, Ph8.0/10 mM EDTA/1% SDS containing same protease inhibitors). After nuclear lysis, chromatin were fragmented to an average size of 300–400 bp with a Branson sonifier 250 with a microtip and cooling on ice. Cell debris was cleared by centrifugation and ChIP library preparation and adaptor ligation was performed using Illumina ChIP-sequencing sample preparation kit according to manufacturer protocol. ChIP-sequencing was performed at University of Washington sequencing core (St. Louis, MO). The data was analyzed using CLC Bio software (Qiagen Inc Germantown, MD).

## Acknowledgements

We are grateful to Enrique A Mesri (Miller school of Medicine, University of Miami), for kindly providing pCEFL-vGPCR construct. This work was supported by the National Cancer Institute at the National Institutes of Health under award numbers P30-CA016520, P01-CA174439, U54-CA190158, R01-CA171979 and R01-CA244074 (to ESR). The funders had no role in study design, data collection, and analysis, decision to publish, or preparation of the manuscript.

## Additional information

### Funding

| Funder | Grant reference number | Author |
| --- | --- | --- |
| National Cancer Institute | P30-CA016520 | Erle S Robertson |
| National Cancer Institute | P01-CA174439 | Erle S Robertson |
| National Cancer Institute | U54-CA190158 | Erle S Robertson |
| National Cancer Institute | R01-CA171979 | Erle S Robertson |
| National Cancer Institute | R01-CA244074 | Erle S Robertson |

The funders had no role in study design, data collection and interpretation, or the decision to submit the work for publication.

### Author contributions

Rajnish Kumar Singh, Conceptualization, Data curation, Formal analysis, Validation, Investigation, Visualization, Methodology, Writing - original draft, Writing - review and editing; Yonggang Pei, Kunfeng Sun, Validation, Investigation, Methodology; Dipayan Bose, Zachary L Lamplugh, Formal analysis, Validation, Investigation, Methodology; Yan Yuan, Formal analysis, Supervision, Investigation, Writing - review and editing; Paul Lieberman, Resources, Formal analysis, Supervision, Investigation, Writing - review and editing; Jianxin You, Software, Formal analysis, Supervision, Investigation, Writing - original draft, Writing - review and editing; Erle S Robertson, Conceptualization, Resources, Data curation, Software, Formal analysis, Supervision, Funding acquisition, Investigation, Visualization, Writing - original draft, Project administration, Writing - review and editing

### Author ORCIDs

Rajnish Kumar Singh https://orcid.org/0000-0001-7414-1170
Yonggang Pei https://orcid.org/0000-0001-7296-8772

Dipayan Bose (iD) https://orcid.org/0000-0003-4789-2838
Zachary L Lamplugh (iD) http://orcid.org/0000-0003-3442-0591
Kunfeng Sun (iD) http://orcid.org/0000-0002-3874-8909
Erle S Robertson (iD) https://orcid.org/0000-0002-6088-2979

## Decision letter and Author response
Decision letter https://doi.org/10.7554/eLife.57436.sa1
Author response https://doi.org/10.7554/eLife.57436.sa2

## Additional files

### Supplementary files
• Supplementary file 1. List of primers used to amplify various HREs constructs within vCyclin promoter and for the real-time PCR.
• Supplementary file 2. HIF1α-binding sites on the KSHV genome in BC3 cells grown under hypoxic conditions.
• Supplementary file 3. HIF1α-binding sites on the KSHV genome in BC3 cells grown under normoxic conditions.
• Supplementary file 4. HIF1α-binding sites on the KSHV genome in infected PBMCs grown under hypoxic conditions.
• Supplementary file 5. HIF1α-binding sites on the KSHV genome in infected PBMCs grown under normoxic conditions.
• Transparent reporting form

### Data availability
The ChIP sequencing data has been submitted to GEO with accession number GSE149401. All data generated and analysed in this study are included in the manuscript and supporting files. Source data files has been provided for Figures: 1D, 2B, 2C, 2D, 6B and 6C. Also, source data files has been provided for supplementary Figure 1-figure supplement 1, Figure2-figure supplement 1, and Figure 4-figure supplement 2.

The following dataset was generated:

| Author(s) | Year | Dataset title | Dataset URL | Database and Identifier |
|---|---|---|---|---|
| Robertson ES, singh RK | 2020 | HIF1α enrichment on KSHV genome in normoxia and hypoxia | https://www.ncbi.nlm.nih.gov/geo/query/acc.cgi?acc=GSE149401 | NCBI Gene Expression Omnibus, GSE149401 |

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
