## [Decision Letter]

**Acceptance summary:**

This work shows how the oncogenic virus Kaposi's sarcoma-associated herpesvirus (KSHV) bypasses cellular suppression of viral genome replication under low oxygen conditions. The authors observe that the KSHV protein v-cyclin targets the cellular factor HIF1α for degradation when oxygen levels are low, and thereby fosters replication of the KSHV genome. This work is yet another example how viruses manipulate their host to ensure the establishment of chronic and lifelong infections.

**Decision letter after peer review:**

Thank you for submitting your article "KSHV-encoded vCyclin can modulate HIF1α levels to promote DNA replication in hypoxia" for consideration by *eLife*. Your article has been reviewed by 3 peer reviewers, and the evaluation has been overseen by a Reviewing Editor and Päivi Ojala as the Senior Editor. The reviewers have opted to remain anonymous.

The reviewers have discussed the reviews with one another and the Reviewing Editor has drafted this decision to help you prepare a revised submission.

Summary:

The manuscript by Singh et al., investigates how Kaposi's sarcoma-associated herpesvirus (KSHV) bypasses HIF1α-mediated repression of DNA replication in KSHV-infected cells in hypoxic conditions. The authors observe that the KSHV protein v-cyclin targets HIF1α for degradation under hypoxia. They show that a knockdown of vCyclin results in suppression of KSHV genome replication as well as anchorage-dependent and anchorage-independent growth.

Essential revisions:

The reviewers all agreed that the manuscript has potential and describes relevant findings, but that several experiments are needed to substantiate them. In addition, introduction, results and discussion should be described much more clearly and require major editing, particularly with regard to the hypotheses, interpretation, and conclusions.

Please address all points listed below that were raised by the three reviewers:

1. The results in Figure 1 are not sufficient to support the main statement of this section that the virus restricts the transcriptional activity of Hif1a.

Figure 1A/B show representative cases of nucleotide analog incorporation (1B) and, in Figure 1C, a representative Hif1a western blot is shown. It would be necessary to show the quantification of all flow cytometry experiments in 1B (% cells) and western blot quantifications in 1C.

Line 159, the authors state that "the hypoxic BJAB-KSHV cells showed no significant increase, but a lower level of expression as compared to BJAB cells in hypoxia (Figure 1D). In this figure, the authors show rtPCR expression of Hif1a targets. It is not clear when the rtPCR experiments were performed. If performed at 36h hours, the decreased fold changes in Figure 1D could simply be explained by the different levels of Hif1a. It is necessary to show the Hif1a target expression at several time points including 36h, when HIF1α levels decrease in hypoxic infected cells.

All the targets reach similar levels of expression in both hypoxic conditions. It would be important to control for the different levels of Hif1a in infected compared to non-infected normoxic cells. The lower fold changes could be due to the high saturating levels of Hif1a.

Based on this, the authors should adjust the title and the conclusions of this first Results section.

2. Figure 4C shows no expression of vCyclin in stable cells. It would be very useful to see the temporal expression of vCyclin to understand the degradation dynamics. From this panel only, it is not clear whether the stable cells express vCyclin. In Figure 4D, in the presence of the proteasomal inhibitor, the levels of vCyclin are reduced. The authors should comment on the implications for the conclusions. Also, intermediate concentrations of the proteasomal inhibitor appear to have higher HIF1α levels compared to the highest concentration.

3. The authors should include quantitative estimates for all western blots for at least three replicates.

4. The authors should provide more details on the results from Figure 4 in the main text. Are the different observations due to different conditions? Also, this figure needs labelling of the different conditions.

5. The section titled "KSHV infection rescues HIF1α-mediated negative regulation of DNA replication" does not show rescue of DNA replication. The results only confirm with ChipSeq the binding of HIF1α on the viral genome. The title needs to be revised.

6. The section titled "Knock-down of vCyclin in KSHV positive cells attenuated KSHV-mediated bypass of hypoxia induced DNA replication and proliferation." should be revised. The authors likely refer to hypoxia-repressed DNA replication. Also, a large part of this section focuses on methodological details rather than the results.

7. The discussion largely focuses on the results of the manuscript. It would be interesting to put the results in the context of what is known in the literature, what other mechanisms could be implicated, the implications of these findings for cancer biology and further research.

8. Could the authors hypothesize why HIF1a levels remain elevated at all timepoints prior to 36 hours? Did the authors check after the 36 hour time point? Please provide data.

9. Figure 1. At what time point does the RT-qPCR in Figure 1D correspond to? Judging by the western blot in Figure 1C HIF1a appears to steadily increase in KSHV-infected cells (however there is no quantification so it's hard to tell for sure). Are HIF1a-target genes responsive to this increased amount of HIF1a early during hypoxia? (assuming there is a steady increase-the quantification would help clarify this).

10. Is the hypoxia induced degradation of HIF1a also observed in naturally infected BC3s (or other PEL lines)?

11. Figure 4. Lysosomal-dependent degradation of HIF1a has previously been described as well as the ability of chloroquine to increase HIF1a stability. Therefore, the specificity of the effects described in Figure 4D are unclear. The effects of chloroquine on HIF1a levels in the absence of vCyclin should be tested and shown.

12. Moreover, CDK2, which has been demonstrated to interact with vCyclin, has previously been shown to mediate the lysosomal-dependent degradation of HIF1a. This raises the question whether the vCyclin-dependent decay of the HIF1a is mediated through CDK2-dependent lysosomal-degradation pathway. Does vCyclin require CDK2 for HIF1a decay? Perhaps vCyclin is high-jacking/enhancing this mechanism of HIF1a decay.

13. Figure 6. It is not clear if the phenotypes described in Figure 6 are related to HIF1a or to other functions of vCyclin. Are HIF1a levels increased upon knockdown of vCyclin? Does overexpression of HIF1a phenocopy the effects of knocking down vCyclin?

14. Tying this all back to the observed degradation of HIF1a as initially described in Figure 1C. What is the expression profile of vCyclin in hypoxia and does this correlate with HIF1a decay in infected cells?

---

## [Author Response]

Essential revisions:The reviewers all agreed that the manuscript has potential and describes relevant findings, but that several experiments are needed to substantiate them. In addition, introduction, results and discussion should be described much more clearly and require major editing, particularly with regard to the hypotheses, interpretation, and conclusions.Please address all points listed below that were raised by the three reviewers:1. The results in Figure 1 are not sufficient to support the main statement of this section that the virus restricts the transcriptional activity of Hif1a.Figure 1A/B show representative cases of nucleotide analog incorporation (1B) and, in Figure 1C, a representative Hif1a western blot is shown. It would be necessary to show the quantification of all flow cytometry experiments in 1B (% cells) and western blot quantifications in 1C.Line 159, the authors state that "the hypoxic BJAB-KSHV cells showed no significant increase, but a lower level of expression as compared to BJAB cells in hypoxia (Figure 1D). In this figure, the authors show rtPCR expression of Hif1a targets. It is not clear when the rtPCR experiments were performed. If performed at 36h hours, the decreased fold changes in Figure 1D could simply be explained by the different levels of Hif1a. It is necessary to show the Hif1a target expression at several time points including 36h, when HIF1α levels decrease in hypoxic infected cells.All the targets reach similar levels of expression in both hypoxic conditions. It would be important to control for the different levels of Hif1a in infected compared to non-infected normoxic cells. The lower fold changes could be due to the high saturating levels of Hif1a.Based on this, the authors should adjust the title and the conclusions of this first Results section.

Thank you very much for the important comment. In the revised manuscript, we now provide the quantitative results of flow cytometry, and the western blots for three replicates. Regarding the results of the real-time PCR experiment shown in Figure 1D, we now clearly include in the text the time point at which the RNA was collected, and the assay performed. The main aim of the experiment shown in Figure 1D was to further validate the observation where we saw low HIF1α levels at the 36-hour time point. The observation of low HIF1a levels at the 36-hour time point (at protein level) and confirmation of the relatively less prominent transcriptional activity of HIF1α at this time point helped us to hypothesize a role for KSHV-encoded factor(s) in this interesting phenomenon. Finally, we were able to elucidate the finding that KSHV-encoded vCyclin interacted and mediated lysosomal degradation of HIF1α, which ultimately strengthened our initial observation and the conclusions of the manuscript. Further, as suggested by the reviewer, we now provide data for expression of HIF1α targets at several time points including that at 36h (Figure 1-figre supplement 1). The data clearly demonstrated this unexpected activity of HIF1α in KSHV positive cells grown under hypoxic conditions. We also agree with reviewer that the lower fold changes of HIF1α targets in BJAB-KSHV cells grown under hypoxic conditions could be due to high saturating levels of HIF1α, which is true as evident from the literature, and new experiments now added in the revised manuscript (Figure 1-figre supplement 1). Nevertheless, the data showed in the proceeding result sections clearly suggests that vCyclin physically interacted and mediated degradation of HIF1α, which is our main conclusion from this work. These findings now allow us to understand how KSHV can bypass HIF1α mediated repression of DNA replication and can lead to reactivation of the virus under hypoxic conditions. The real-time PCR experiment for HIF1α targets at different time points, in addition to other experiments now shown in the revised manuscript clearly support our observations and conclusion. However, based on reviewers’ comments, we have adjusted the title and the conclusion of first result section.

2. Figure 4C shows no expression of vCyclin in stable cells. It would be very useful to see the temporal expression of vCyclin to understand the degradation dynamics. From this panel only, it is not clear whether the stable cells express vCyclin. In Figure 4D, in the presence of the proteasomal inhibitor, the levels of vCyclin are reduced. The authors should comment on the implications for the conclusions. Also, intermediate concentrations of the proteasomal inhibitor appear to have higher HIF1α levels compared to the highest concentration.

Thank you for the comment. As we have shown by fluorescent microscopy, the transiently transfected cells have difficulty to stably express vCyclin (Figure 4C). In fact, the expression of transfected vCyclin (GFP-tagged) showed a time-dependent continuous fading which ultimately results in little or no detection of fluorescent microscopy-based detection of GFP (tagged to vCyclin). Interestingly, these cells also grow well when selected in the presence of antibiotics. As per suggestion of reviewer, we have now provided data for temporal degradation of vCyclin in transfected cells. Here, we compared the expression level of vCyclin at 24-hour intervals post 24-hours of transfection. As can be seen in (Figure 4-figre supplement 1), the GFP fluorescence continuously faded away and was completely diminished after nearly one week post-transfection. It is important to note that transfected mock GFP plasmid retained their fluorescence throughout the studied period, as well in stably transfected cells.

In figure 4D, we did observe that in the presence of proteasomal inhibitor MG132, there was lower expression of vCyclin. As we mentioned in the manuscript, vCyclin interacts with HIF1α and translocated it to lysosomal compartment. The low levels of vCyclin in proteasomal inhibitor treated cells could be due to lysosomal degradation of vCyclin itself. In addition, the phenomenon of lysosomal degradation is absent in MG132 treated cells, hence we observed higher levels of both HIF1α as well as vCyclin. It is important to note that equal amounts of GFP-vCyclin or HIF1α plasmids were used in the transfection experiment. A less possible explanation could be due to the inhibitory effect of MG132 at the transcriptional or translational level for vCyclin. Additionally, we cannot completely rule out higher cytotoxic effects of MG132 as compared to Chloroquine. We know that proteasomal degradation is an alternate pathway for HIF1α degradation through vHL in normoxic conditions. Therefore, it is expected that there will be some level of protection of HIF1α levels in the presence of MG132. This may be the reason for higher levels of HIF1α at a higher concentration. Further, at the highest concentration of MG132 used, the lowering in the levels of HIF1α could be due to the possible cytotoxic effect of MG132. Hence, optimization of MG132 concentrations is always recommended in all experiments to minimize any cytotoxic effects of the drug.

3. The authors should include quantitative estimates for all western blots for at least three replicates.

Thank you for the comment. In the previous version of manuscript, we did already provide quantitation of Figure 4A, Figure 4B and (Figure 4-figure supplement 1). In the revised manuscript, we now provide additional quantitation of the western blots where they are needed. These include Figures 1C, Figure 4D, and Figure 4-figure supplement 1. Western blots such as those given in Figure 2B, Figure 3C, Figure 3D, Figure 3E, Figure 5B and Figure 5C are used only qualitative representation such as for showing expression of transfected constructs or protein-protein interaction and therefore we did not believe that quantitation was not necessary or required as there were no scientific reasons to include them for these blots.

4. The authors should provide more details on the results from Figure 4 in the main text. Are the different observations due to different conditions? Also, this figure needs labelling of the different conditions.

Thank you for the comment. We have now provided more details for the results seen in Figure 4 described in the main text of revised manuscript. The different observations mentioned in the Results section for Figure 4, especially that of Figure 4C is due to temporal localization of HIF1α in the different cells shown. Representative images were shown for different cells based on their localization of the specific proteins, whether they are cytosolic, lysosomal, or in cyto-lysosomal compartments. Further, we have now provided clear labelling in the revised manuscript.

5. The section titled "KSHV infection rescues HIF1α-mediated negative regulation of DNA replication" does not show rescue of DNA replication. The results only confirm with ChipSeq the binding of HIF1α on the viral genome. The title needs to be revised.

Thank you very much for the comments. The main aim of the experiment was to show that in hypoxic conditions, enrichment of HIF1α is not centered towards the origin of lytic replication, and this observation provides evidence that the DNA is not undergoing lytic replication. Also, enrichment of HIF1α in these regions can have a negative effect on KSHV replication through its well-known role in replication silencing through its interaction with a number of replication-associated proteins. Finally, based on the suggestion of the reviewer, we have modified the title of this section according to the results that we obtained.

6. The section titled "Knock-down of vCyclin in KSHV positive cells attenuated KSHV-mediated bypass of hypoxia induced DNA replication and proliferation." should be revised. The authors likely refer to hypoxia-repressed DNA replication. Also, a large part of this section focuses on methodological details rather than the results.

Thank you very much for the comments. We have modified the title and content of this section according to the results in the revised manuscript. We also thank to the reviewer for pointing out the sentence where we actually wanted to refer to “Hypoxia-repressed DNA replication”. Further, we have now removed the unnecessary methodological details in the text part of this section and focused our efforts on describing the results in more detail in the revised version of the manuscript.

7. The discussion largely focuses on the results of the manuscript. It would be interesting to put the results in the context of what is known in the literature, what other mechanisms could be implicated, the implications of these findings for cancer biology and further research.

Thank you very much for the valuable comment. In the revised manuscript, we have put greater care in developing this section with implications for cancer biology and additional research as suggested. In addition to discussing the results of our work, we have now discussed in more detail how these findings fit in the context of the existing literature in the broader context of cancer biology. We have also discussed the possibilities of other mechanisms which can be implicated in the context to existing knowledge as well as insights into future research directions.

8. Could the authors hypothesize why HIF1a levels remain elevated at all timepoints prior to 36 hours? Did the authors check after the 36 hour time point? Please provide data.

Thank you for the comment. Elevated levels of HIF1α is a common observation for many cancers, KSHV-positive, and other oncogenic virus-infected cells. Particularly, in KSHV-infected cells, KSHV-encoded LANA and vGPCR are responsible for this elevated level of HIF1α. LANA is known to inhibit vHL dependent proteasomal degradation of HIF1α. vGPCR is known to up-regulate expression of HIF1α by working on MAP Kinase/p38 pathway. Interestingly, both factors (LANA and vGPCR) are under hypoxic control where expression of these proteins get up-regulated in hypoxic conditions. The up-regulation of LANA and vGPCR is due to HIF1α mediated transactivation of the hypoxia responsive elements within the promoter region. Therefore, KSHV-encoded LANA and vGPCR creates a positive feedback loop for maintaining high levels of HIF1α. Furthermore, since LANA and vGPCR are also under hypoxic control, a very high level of HIF1α is expected in KSHV-positive cells grown under hypoxic conditions. In addition, based on known inhibitory role of HIF1α on cell cycle and DNA replication, it was necessary to question how KSHV-positive cells kept growing in hypoxic conditions as well as been activated for lytic replication in hypoxia. Based on our hypothesis that KSHV bypasses HIF1α-mediated repression of DNA replication, our observation of lower levels of HIF1α at 36 hours led us to confirm this observation with the transcriptional signature of HIF1α. Interestingly, we found that KSHV-encoded vCyclin can degrade HIF1α through the lysosomal pathway. Putting these together to answer the question as to why HIF1α levels remain elevated at all timepoints prior to 36 hours, we think this is the time period required to accumulate sufficient vCyclin at the protein levels, and translocation of interacted HIF1α to lysosomal compartments followed by its degradation, we consider this phenomenon as the vCyclin dominated-HIF1α depleted phase. In this phase HIF1α mediated repression of DNA replication will be released. It is also important to note that vCyclin itself undergoes degradation in the lysosomal compartment, which would lead to increased levels of HIF1α-mediated by LANA and vGPCR. This phase we referred to as vCyclin depleted-HIF1α dominated phase. In this phase, we hypothesize that cells will allow HIF1α mediated transcription to accumulate resources needed for another cycle of replication. With respect to the rest of the comment, we checked levels of HIF1α beyond the 36-hours time point, as well as its transcriptional activity. We observed a marginal increase in both its protein levels, as well as its transcriptional activity up to 60 hours. These data are now provided in the revised manuscript (Figure 1—figure supplement 1). We cannot extend the hypoxia-based experiment for a longer time period as this is in the closed cell culture condition, and so the cells will completely consume the glucose in the media and release high amounts of lactate in the surrounding medium. Further, the effect of lytic replication of virus in hypoxic conditions, on the viability of infected cells, is another limiting factor for longer time point based experiments. An in vivo experimentation will be great way to see these two different phases of cells in hypoxic conditions. Alternatively, a system can be developed where we can continuously supply nutrients to the cells growing under hypoxic conditions without disturbing the hypoxic microenvironment. These all urge a new elaborated research project which we would like to execute in future.

9. Figure 1. At what time point does the RT-qPCR in Figure 1D correspond to? Judging by the western blot in Figure 1C HIF1a appears to steadily increase in KSHV-infected cells (however there is no quantification so it's hard to tell for sure). Are HIF1a-target genes responsive to this increased amount of HIF1a early during hypoxia? (assuming there is a steady increase-the quantification would help clarify this).

Thank you very much for the important comment. The RT-PCR experiment shown in figure 1D was performed at 36 hours of treatment. We have now clearly mentioned this in the revised manuscript. The aim for taking this time point was to confirm the observation of western blot analysis where we found low levels of HIF1α. Further based on the major comment 1 by the reviewer, we performed real-time experiments at various time points. The results showed that the fold change expression of HIF1α target in BJAB-KSHV cells were less prominent mainly at higher time point when compared to their KSHV-negative counterpart BJAB cells. Hence, we can conclude that HIF1α-target genes are responsive to the increased amount of HIF1α early during hypoxia. Further, as per reviewers’ suggestion, we now provide data showing quantitation for the Western blot experiments in the revised manuscript.

10. Is the hypoxia induced degradation of HIF1a also observed in naturally infected BC3s (or other PEL lines)?

Thank you for the comment. In the present study, we observed that in naturally infected cells such as BC3, the vCyclin interacts with HIF1α and this leads to translocation to the cytosolic compartment during hypoxia (Figure 4C). We therefore hypothesize that hypoxia can induce degradation of HIF1α in naturally infected cells. In the revised manuscript, we now provide data which clearly suggests that HIF1α is degraded in naturally infected cells grown under hypoxia. The results also showed that knock down of vCyclin can lead to higher expression of HIF1α in the cells (Figure 4—figure supplement 2 and Figure 6—figure supplement 1).

11. Figure 4. Lysosomal-dependent degradation of HIF1a has previously been described as well as the ability of chloroquine to increase HIF1a stability. Therefore, the specificity of the effects described in Figure 4D are unclear. The effects of chloroquine on HIF1a levels in the absence of vCyclin should be tested and shown.

Thank you for the comment. We agree with the reviewer that lysosomal degradation of HIF1α was described previously, and is mediated by CDK2. However, CDK2-mediated lysosomal degradation of HIF1α was observed independent of KSHV, especially in normoxia. It would be important to note that CDK2 itself undergoes degradation in hypoxia and that KSHV infection can stabilize CDK2 from hypoxia mediated degradation (Singh et.al., 2019; PLoS Pathogens). This emphasizes the high level of importance of vCyclin-induced lysosomal degradation of HIF1α in KSHV infected cells. The specificity of vCyclin-mediated degradation of HIF1α was confirmed through several experiments shown in Figure 4D-E, and Supplementary Figure 4C. We used MG132 in the experiment showed in Figure 4D, a proteasomal inhibitor which is also known to reduce degradation of HIF1α. This can act as another control for the chloroquine treatment experiment. In this experiment, only chloroquine was able to rescue vCyclin mediated lysosomal degradation of HIF1α in a dose dependent manner. However, based on the reviewer’s suggestion in this query, and other queries as to whether hypoxia induces degradation of HIF1α in other naturally infected cells, or whether HIF1α level increases after knock down of vCyclin, we now provide additional experiments where we used knock down conditions of vCyclin with or without chloroquine treatment. These results provided in the revised manuscript responds to all these specific queries (Figure 4—figure supplement 2 and Figure 6—figure supplement 1).

12. Moreover, CDK2, which has been demonstrated to interact with vCyclin, has previously been shown to mediate the lysosomal-dependent degradation of HIF1a. This raises the question whether the vCyclin-dependent decay of the HIF1a is mediated through CDK2-dependent lysosomal-degradation pathway. Does vCyclin require CDK2 for HIF1a decay? Perhaps vCyclin is high-jacking/enhancing this mechanism of HIF1a decay.

Thank you very much such an important comment. This CDK2-vCyclin-HIF1α axis imposes a very interesting scenario when exogenous hypoxia is induced in a KSHV infected cell. In non-infected cells, HIF1α was known to be regulated primarily by vHL-mediated proteasomal degradation in normoxic cells. Moreover, CDK2-mediated lysosomal degradation of HIF1α was recently reported. We recently reported an interesting phenomenon where in KSHV-negative cells, CDK2 undergoes degradation under hypoxic conditions. Further, KSHV-infection can rescue CDK2 degradation, but requires HIF1α (Singh et al., 2019; PLoS Pathogens). This suggests that, CDK2-mediated lysosomal degradation of HIF1α will not happen if vCyclin degrades HIF1α under hypoxic conditions. We failed in our attempt to generate a functional shCDK2 cell in the KSHV background, which would have clearly demonstrated CDK2-independent degradation of HIF1α by vCyclin. This is likely to be due to the fact that CDK2 is a critical cell cycle factor. We will peruse this interesting question in more detail in our future studies.

13. Figure 6. It is not clear if the phenotypes described in Figure 6 are related to HIF1a or to other functions of vCyclin. Are HIF1a levels increased upon knockdown of vCyclin? Does overexpression of HIF1a phenocopy the effects of knocking down vCyclin?

Thank you for the valuable comment. We performed new experiments to confirm that the results obtained in Figure 6 is due to vCyclin-mediated restriction of HIF1α functions of replication inhibition. Here we now show that knock-down of vCyclin, results in high levels of HIF1α compared to mock cells when grown under hypoxic conditions. Further, higher expression of observed HIF1α in vCyclin knock down cells were due to inhibition of the lysosomal activity which further confirms its specificity (Figure 4—figure supplement 2).

14. Tying this all back to the observed degradation of HIF1a as initially described in Figure 1C. What is the expression profile of vCyclin in hypoxia and does this correlate with HIF1a decay in infected cells?

Thank you very much for this important query. We are providing results from recently completed experiments to show the expression profile of vCyclin under hypoxic conditions at different time points and the corresponding HIF1a decay on those time points. The results clearly confirms that HIF1α decay is directly proportional to vCyclin levels (Figure 6—figure supplement 1).